# Spatio-temporal associations between deforestation and malaria incidence in Lao PDR

Francois Rerolle[1,2]*, Emily Dantzer[1], Andrew A Lover[3], John M Marshall[4], Bouasy Hongvanthong[5], Hugh JW Sturrock[1,2], Adam Bennett[1,2]

[1]Malaria Elimination Initiative, The Global Health Group, University of California, San Francisco, San Francisco, United States; [2]Department of Epidemiology and Biostatistics, University of California, San Francisco, San Francisco, United States; [3]Department of Biostatistics and Epidemiology, School of Public Health and Health Sciences, University of Massachusetts Amherst, Amherst, United States; [4]Division of Epidemiology and Biostatistics, School of Public Health, University of California, Berkeley, Berkeley, United States; [5]Center for Malariology, Parasitology and Entomology, Ministry of Health, Vientiane, Lao People's Democratic Republic

**Abstract** As countries in the Greater Mekong Sub-region (GMS) increasingly focus their malaria control and elimination efforts on reducing forest-related transmission, greater understanding of the relationship between deforestation and malaria incidence will be essential for programs to assess and meet their 2030 elimination goals. Leveraging village-level health facility surveillance data and forest cover data in a spatio-temporal modeling framework, we found evidence that deforestation is associated with short-term increases, but long-term decreases confirmed malaria case incidence in Lao People's Democratic Republic (Lao PDR). We identified strong associations with deforestation measured within 30 km of villages but not with deforestation in the near (10 km) and immediate (1 km) vicinity. Results appear driven by deforestation in densely forested areas and were more pronounced for infections with *Plasmodium falciparum* (*P. falciparum*) than for *Plasmodium vivax* (*P. vivax*). These findings highlight the influence of forest activities on malaria transmission in the GMS.

*For correspondence:
francois.rerolle@ucsf.edu

Competing interests: The authors declare that no competing interests exist.

## Introduction

Engaging in forest activities, such as logging, hunting, or spending the night in the forest, is considered a primary risk factor for malaria infection in the Greater Mekong Sub-region (GMS) (*Chaveepojnkamjorn and Pichainarong, 2004*; *Das et al., 2004*; *Lansang et al., 1997*; *Erhart et al., 2005*; *Trung et al., 2004*) with recent outbreaks attributed to deforestation activities (*Lyttleton, 2016*). This is most likely the result of increased human exposure to the forest dwelling *Anopheles dirus* and *Anopheles minimus*, the most efficient and widespread malaria vectors in the GMS (*Obsomer et al., 2007*; *Obsomer et al., 2013*). However, deforestation may also alter this 'forest malaria' (*Sharma and Kondrashin, 1991*) ecosystem and eventually reduce malaria incidence, as is generally accepted to be the case in Southeast Asia (*Guerra et al., 2006*). Several previous studies have assessed the relationship between deforestation and malaria, but the majority focused outside of the GMS, most notably in the Amazonian forest (*Wayant et al., 2010*; *Olson et al., 2010*; *Hahn et al., 2014a*; *Valle and Clark, 2013*; *Terrazas et al., 2015*) where the evidence has been met with conflicting interpretations (*Tucker Lima et al., 2017*). As national malaria programs across the GMS target forest-going populations for prevention and treatment efforts (*Guyant et al., 2015*; *WHO, 2016*), improved understanding of the relationship between deforestation and malaria is

**eLife digest** Biting mosquitos spread the malaria parasite to humans. Along the Mekong River in Southeast Asia, spending time in the surrounding forest increases a person's risk of malaria. This has led to a debate about whether deforestation in this area, which is called the Greater Mekong Sub-region (GMS), will increase or decrease malaria transmission. The answer to the debate is not clear because some malaria-transmitting mosquitos thrive in heavily forested areas, in particular in the GMS, while others prefer less forested areas.

Scientists studying malaria in the Amazon in South America suspect that malaria transmission increases shortly after deforestation but decreases six to eight years later. Some studies have tested this 'frontier malaria' theory but the results have been conflicting. Fewer studies have tested this theory in Southeast Asia. But deforestation has been blamed for recent malaria outbreaks in the GMS.

Using data on malaria testing and forest cover in the GMS, Rerolle et al. show that deforestation around villages increases malaria transmission in the first two years and decreases malaria rates later. This trend was driven mostly by a type of malaria called *Plasmodium falciparum* and was less strong for *Plasmodium vivax*. The location of deforested areas also mattered. Deforestation within one to 10 kilometer of villages did not affect malaria rates. Deforestation further away in about a 30 kilometer radius did affect malaria transmission. Rerolle et al. suggest this may be because villagers have to spend longer times trekking through forests to hunt or harvest wood when the wider area is deforested.

Currently, National Malaria Control Programs in the GMS focus their efforts on reducing forest-related transmission. This study strengthens the evidence supporting this approach. The results also suggest that different malaria elimination strategies may be necessary for different types of malaria parasite. Using this new information could help malaria control programs better target resources or educate villagers on how to protect themselves. The innovative methods used by Rerolle et al. reveal a more complex role of deforestation in malaria transmission and may inspire other scientists to think more carefully about environmental drivers of malaria.

critical for programs to assess and meet national 2030 malaria elimination goals (*World Health Organization, 2016*; *WHO, 2018*).

In the Amazon, the 'frontier malaria' hypothesis (*Sawyer, 1988*) posits that malaria temporarily increases following deforestation efforts to open a human settlement area in the forest. Subsequently, after approximately 6–8 years, settlements become more urbanized and isolated from the surrounding forest, and less suitable for malaria vectors, resulting in reduced malaria transmission (*de Castro et al., 2006*). Recent work has challenged this hypothesis, however, and found that some older settlements were also likely to have high malaria incidence (*Ilacqua et al., 2018*), highlighting the importance of assessing the relationship between deforestation and malaria at different spatio-temporal scales (*Singer and de Castro, 2006*).

A recent review of the literature on deforestation and malaria in the Amazon (*Tucker Lima et al., 2017*) recommended the integration of multiple socio-economic, demographic and ecological mechanisms to disentangle the relationship between deforestation and malaria. The complexity of land-use changes driving deforestation such as urbanization, agriculture, irrigation or resource mining can alter the environment in different ways. For example, deforestation in the Amazon has been shown to increase mosquitoes' larval habitat (*Vittor et al., 2009*) through the creation of areas with abundant sunlight and pooling water, resulting in increased human biting activity (*Vittor et al., 2006*). Alternatively, immigration and rapid population movements, stirring human-vector interactions, are other mechanisms affecting malaria transmission in frontier areas (*Moreno et al., 2007*). A modeling study (*Baeza et al., 2017*) showed that the temporal pattern of increased incidence followed by a decrease can vary depending on ecological and socio-economic parameters in frontiers areas.

The importance of addressing complex confounding structures influencing the relationship between deforestation and malaria was also highlighted by *Bauhoff and Busch, 2020*. Variables such as temperature (*Beck-Johnson et al., 2013*; *Mordecai et al., 2013*), precipitation (*Parham and Michael, 2010*; *Parham et al., 2012*), or seasonality *Hay et al., 1998* are known environmental

predictors of malaria, although the spatio-temporal scale of those effects often varies across different areas (*Teklehaimanot et al., 2004*). Furthermore, remote areas may experience higher malaria rates because of poor access to public health services, but also have denser forest cover or lower deforestation rates (*Busch and Ferretti-Gallon, 2017*). Finally, forest-going populations in the GMS are also at higher risk for malaria (*Parker et al., 2017*) due to poor adherence to protective measures against mosquitoes such as insecticide-treated bed nets (ITNs) or long-lasting insecticidal hammocks (LLIHs) (*Peeters Grietens et al., 2010*; *WHO, 2016*) and inadequate access to treatment (*Guyant et al., 2015*).

*Bauhoff and Busch, 2020* identified only 10 empirical studies that assessed the relationship between deforestation and malaria with appropriate adjustments for confounding. Of these, seven reported a positive association (*Austin et al., 2017*; *Wayant et al., 2010*; *Olson et al., 2010*; *Terrazas et al., 2015*; *Pattanayak et al., 2010*; *Garg, 2015*; *Fornace et al., 2016*), two did not find any associations (*Bauhoff and Busch, 2020*; *Hahn et al., 2014a*), and one disputed study found a negative association (*Valle and Clark, 2013*; *Hahn et al., 2014b*; *Valle, 2014*). Most recently, a study found deforestation to increase malaria risk and malaria to decrease deforestation activities in the Amazon, using an instrumental variable analysis to disentangle any reverse causality loop (*MacDonald and Mordecai, 2019*). However, only half of the above-mentioned studies used high-resolution forest data, with most studies using spatially aggregated data and exploring only a limited range of spatial and temporal scales. Only three of these studies were conducted in Southeast Asia (*Pattanayak et al., 2010*; *Garg, 2015*; *Fornace et al., 2016*), and none in the GMS. Importantly, all three found that malaria increases after deforestation, but all had limitations. The two studies in Indonesia *Pattanayak et al., 2010*; *Garg, 2015* used coarsely aggregated forest data and potentially biased self-reported malaria data. The third study, in Malaysia (*Fornace et al., 2016*), focused on a specific and geographically confined malaria parasite, *Plasmodium knowlesi*, whose primary host is the long-tailed macaque and whose presence in the GMS, where *P. falciparum* and *P. vivax* dominate, is limited.

In this analysis, we examined the relationship between deforestation and malaria incidence by combining high-resolution forest coverage data (*Hansen et al., 2013*) and monthly malaria incidence data from 2013 to 2016 from two separate regions in the GMS: northern Lao People's Democratic Republic (PDR) with very low malaria transmission and southern Lao PDR where *P. falciparum* and *P. vivax* are seasonal. By conducting the analysis at the village level, we were able to explore a wide range of spatial scales (1, 10, and 30 km around villages) that might be relevant in characterizing the relationship between deforestation and malaria. In addition, we leveraged the longitudinal nature of both the incidence data collected and the forest data produced from annual remote sensing imagery (*Hansen et al., 2013*) to explore the most relevant temporal scales. Finally, we considered alternative definitions of deforestation, restricted to areas with at least certain levels of forest cover, to investigate the type of deforestation driving the relationship with malaria.

To date, no prior studies have quantified the relationship between deforestation and malaria incidence in the GMS. Understanding this relationship is especially important in the GMS, where forest-going activities are a main source of income generation (*Lao PDR, 2016*) and malaria clusters in forest-going populations (*World Health Organization, 2016*; *WHO, 2016*). To assess the relationship between deforestation and malaria incidence, we modeled the monthly village-level malaria incidence in two regions of Lao PDR using health facility surveillance data and evaluated the most relevant spatio-temporal scale.

## Results

### Forest and environmental data

*Figure 1* shows the average tree crown cover within 10 km for the year 2016 and the percent area within 10 km that experienced forest loss between 2011 and 2016 in two regions of southern and northern Lao PDR. Overall, the forest cover was denser in the north than in the south and deforestation over this period was higher in the north than in the south. *Appendix 1—figure 6* and *Appendix 1—figure 7* show the distribution of forest and deforestation variables as the temporal scales and spatial scales around study villages were varied. For example, the cumulative percent area within 30 km of a village that experienced forest loss between 2011 and 2016 ranged from 0 to 10% in the

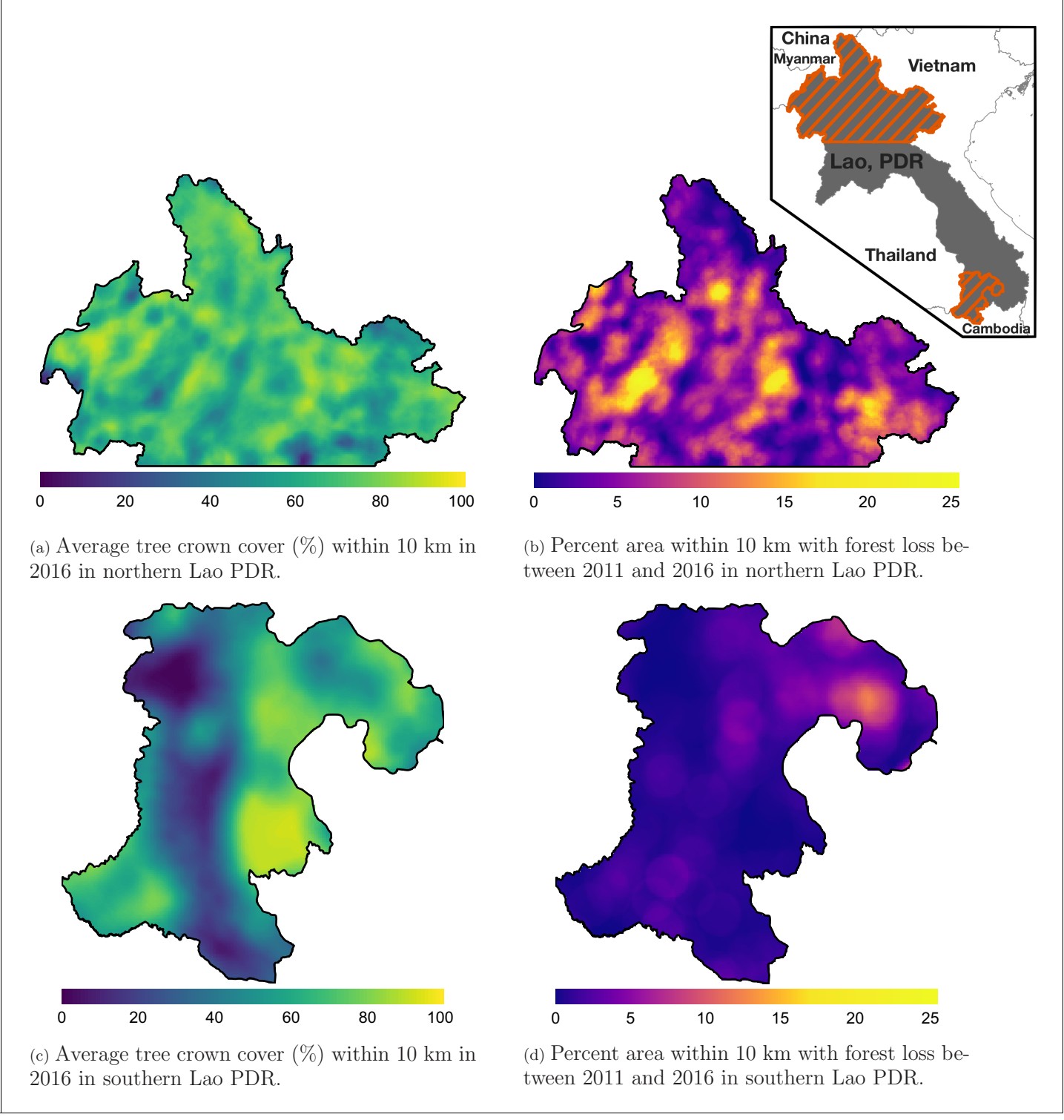

(a) Average tree crown cover (%) within 10 km in 2016 in northern Lao PDR.

(b) Percent area within 10 km with forest loss between 2011 and 2016 in northern Lao PDR.

(c) Average tree crown cover (%) within 10 km in 2016 in southern Lao PDR.

(d) Percent area within 10 km with forest loss between 2011 and 2016 in southern Lao PDR.

**Figure 1.** Average tree crown cover (%) in 2016 (left) and percent area that experienced forest loss between 2011 and 2016 (right) within a 10 km radius in northern (top) and southern (bottom) Lao PDR. See Materials and methods for details on forest and deforestation metrics. Upper right indent maps northern and southern (Champasak province) Lao PDR regions.

north, whereas it rarely exceeded 2.5% in the south. Deforestation rate in 2015 within 10 km of a village was of about 1% in the south and 2.5% in the north. The average tree crown cover increased with increasing buffer radius around villages (1, 10, and 30 km). However, the relationship with the

percent area that experienced forest loss was less clear because both the area that experienced forest loss (numerator) and the area around villages (denominator) increased. *Appendix 1—figure 15* and *Appendix 1—figure 16* show the raw time series of forest cover and percent area with forest loss.

The 284 villages in the north were overall less populated (mean 2015 population size: 498, IQR: [241; 548]) than the 207 villages in the south (2015 mean: 1095, IQR: [584; 1384]), but with some highly populated outliers. As expected, altitude differed substantially between villages of the mountainous northern region (mean: 557 m, IQR: [378; 679]) and the lowlands of the south (mean: 120 m, IQR: [98; 130]). Although both regions exhibited similar seasonal trends in precipitation and temperature, with a rainy season spanning from April to October, villages in the south experienced higher monthly precipitation and temperature than in the north over the study period (*Appendix 1—figure 8*).

## Treatment-seeking data

For villages with an estimated travel time of 0 hr to the closest health facility (same 300 m$^2$ pixel), the predicted probability of seeking treatment for fever was 0.87 (95% CI: [0.79; 0.92]) in the north and 0.78 (95% CI: [0.63; 0.89]) in the south. A 1 hr increase in travel time to the closest health facility was associated with a similar 0.79 (95% CI: [0.55; 1.13]) reduction in the odds of seeking treatment in the north and 0.76 (95% CI: [0.43; 1.34]) in the south, almost reaching statistical significance when pooling data from both regions: 0.77 (95% CI: [0.56; 1.04]). See detailed results in Appendix 1 - S2.

## Malaria case data

### Malaria infections

A total of 63,040 patient records were abstracted from the malaria registries of all public health facilities in four southern districts between October 2013 and October 2016 and 1754 from all health facilities in four northern districts between January 2013 and December 2016.

In the south, 91.2% of the patients in the registries were tested for malaria, of which 78.1% were tested by RDT and 26.2% by microscopy. Overall test positivity was 33.2% for any infection, 16.4% for *P. falciparum* and 18.2% for *P. vivax*. Monthly incidence peaked to about six cases per 1000 people in the 2014 rainy season, eventually decreasing to below one case per 1000 in 2016. Incidence and test positivity were similar between *P. falciparum* and *P. vivax* in the south (*Figure 2*).

In the north, 92.1% of the patients in the registries were tested for malaria, of which 96.3% were tested by RDT and 9.6% by microscopy. Overall test positivity was 23.8% for any infection, 2.8% for *P. falciparum* and 22.5% for *P. vivax*. Monthly malaria incidence in the north was very low, never exceeding 0.3 per 1000 people. Most infections in the north were *P. vivax* cases with only a few seasonal *P. falciparum* cases (*Figure 2*).

*Appendix 1—figure 9* shows the number of patients and cases recorded per month in health facility malaria registries as well as how the smoothed test positivity rates varied over time.

### Socio-demographics

Age, gender, and occupation of patients seeking treatment at health facilities were also recorded in the malaria registries. On average, patients in the south were older than patients in the north with mean age of 28 years and 23 years, respectively. In the north, about half of the patients were male (53.1%), while most patients in the south were male (71.1%). Finally, the vast majority (68.2%) of patients in the south were farmers, whereas only 8% of patients in the north were farmers. Most patients in the north reported being unemployed (41.7%) or a student (31.2%) (*Appendix 1—figure 10*).

### Geo-referencing

Overall, 88.1% of malaria records were matched to one of the 491 villages in study districts. The remaining (11.7% in the south and 17.3% in the north) were removed from the analysis because of ambiguous village names, local nicknames for small villages and dissolving and grouping of villages over time. Test positivity in the south was similar in matched (33.1%) and unmatched (34.2%) records but higher in matched (26.5%) than unmatched (10.5%) records in the north. No substantial difference was found in the distribution of socio-demographic variables available in malaria registries

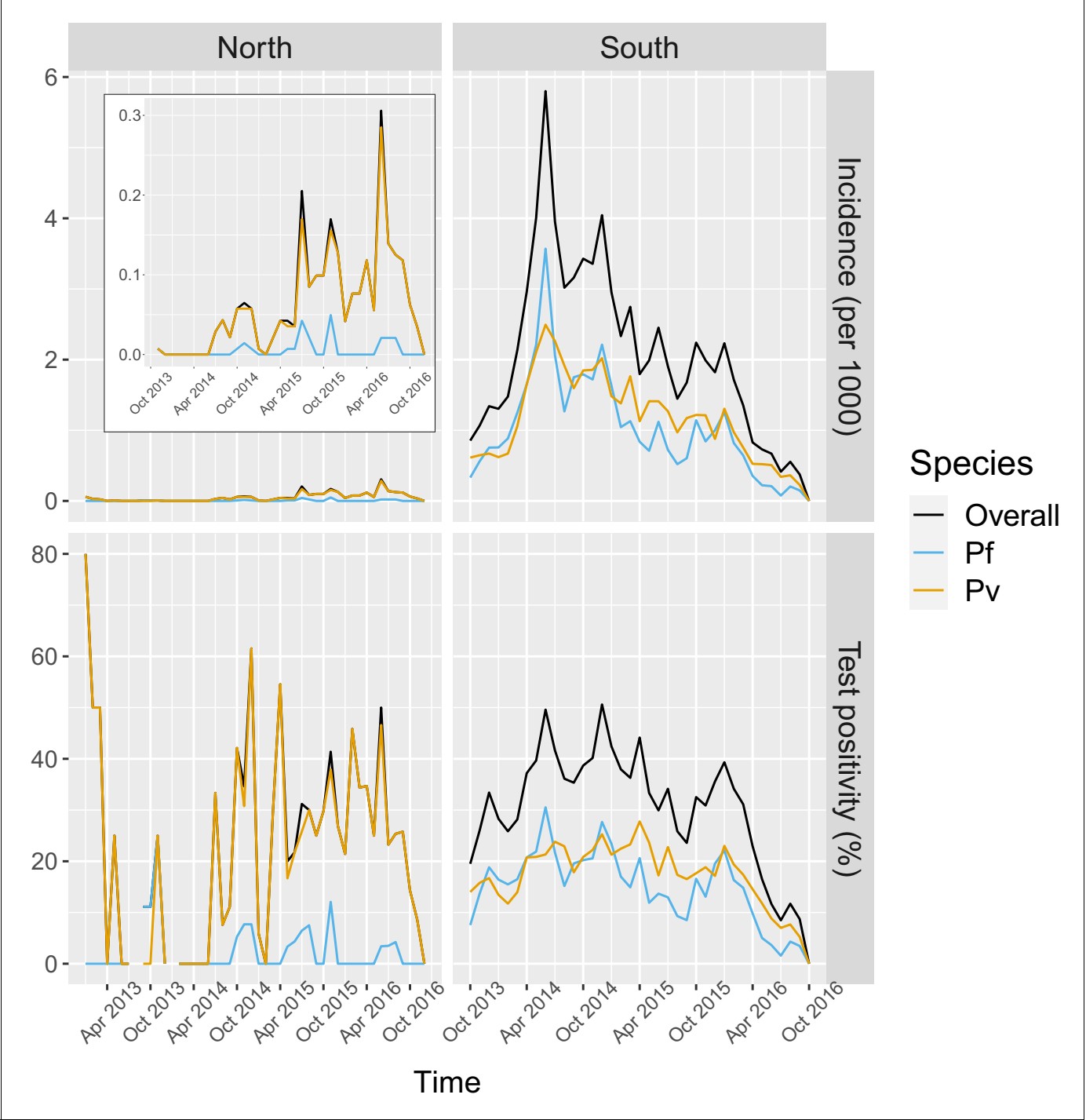

**Figure 2.** Malaria incidence (per 1000) and test positivity (%) over time. Upper left boxed indent zooms in malaria incidence in the North to better show the temporal variation (see y axis for scale).

The online version of this article includes the following source data for figure 2:

**Source data 1.** Data for *Figure 2*.

between matched and unmatched records (*Appendix 1—figure 11*). Fewer than 0.3% of matched malaria records were missing dates and also removed from the analysis.

## Spatio-temporal analysis
### Deforestation

*Table 1* and *Figure 3* show the adjusted incidence rate ratio (IRR) associated with deforestation, measured by a 0.1% increase in the percent area that experienced forest loss, in the previous 1–5 years within 1, 10, and 30 km of villages. Models controlled for various environmental factors and accounted for the probability of seeking treatment and the spatio-temporal structure of the data.

Deforestation within 1 or 10 km of a village was not associated with malaria incidence rate in either the south or the north, regardless of the temporal lag. However, in the south, deforestation within 30 km of a village in the previous 1 and 2 years was associated with higher malaria incidence rates (e.g. 1 year lag, IRR = 1.16, 95% CI: [1.10; 1.22]). In the north, where incidence was much lower, the results were not as clear, but a similar trend was observed with wide confidence intervals compatible with a short term increased risk. On the other hand, deforestation within 30 km of a village in the previous 3, 4, or 5 years was associated with approximately a 5% lower malaria incidence rate both in the south (e.g. 5 year lag, IRR = 0.94, 95% CI: [0.91; 0.97]) and in the north (e.g. 5 year lag, IRR = 0.96, 95% CI: [0.93; 0.98]).

These results suggest deforestation around villages, but not in the near vicinity (1 or 10 km), is associated with higher risk of malaria in the first 2 years and lower risk of malaria beyond. There was stronger evidence of associations with deforestation in the south than in the north.

The IRR effect estimates in *Table 1* assume a linear relationship between deforestation and malaria. *Appendix 1—figure 5* shows a few of these relationships when such linearity isn't assumed in the models. The functional forms reveal that they can be reasonably well summarized linearly, especially in the south. In the north, the functional forms highlight potential non-linearities for long-term temporal lags but come with wide confidence intervals at extreme levels of deforestation.

**Table 1.** IRR between malaria incidence and a 0.1% increase in the area that experienced deforestation within 1, 10, or 30 km (left-right) of a village in the previous 1– 5 years (top-down) in northern and southern Lao PDR.

Adjusted for the probability of seeking treatment, the spatio-temporal structure of the data, the environmental covariates selected in the model and forest cover within 30 km in the year before the deforestation temporal scale considered as well as for malaria incidence in the previous 1 and 2 years. See Materials and methods for details.

| | South | | | North | | |
|---|---|---|---|---|---|---|
| **Time lag** | **Buffer radius** | | | **Buffer radius** | | |
| | 1 km | 10 km | 30 km | 1 km | 10 km | 30 km |
| Previous 1 year | 1 [0.99; 1.01] | 1.01 [0.99; 1.04] | 1.16 [1.10; 1.22] | 1 [1; 1.01] | 1.03 [0.99; 1.06] | 1.01 [0.94; 1.08] |
| Previous 2 years | 1 [0.99; 1.01] | 1 [0.98; 1.01] | 1.08 [1.04; 1.13] | 1 [1; 1.01] | 1.01 [0.99; 1.04] | 0.99 [0.95; 1.03] |
| Previous 3 years | 0.99 [0.99; 1] | 0.98 [0.97; 1] | 0.93 [0.90; 0.97] | 1 [1; 1.01] | 1.01 [0.99; 1.02] | 0.96 [0.94; 0.99] |
| Previous 4 years | 0.99 [0.99; 1] | 0.98 [0.97; 0.99] | 0.94 [0.92; 0.97] | 1 [1; 1.01] | 1 [0.99; 1.02] | 0.97 [0.94; 0.99] |
| Previous 5 years | 1 [0.99; 1] | 0.97 [0.96; 0.99] | 0.94 [0.91; 0.97] | 1 [1; 1.01] | 1.01 [0.99; 1.02] | 0.96 [0.93; 0.98] |

The online version of this article includes the following source data for Table 1:

Source data 1. Data for *Table 1*.

### *P. falciparum* and *P. vivax*

In addition to different overall levels of transmission in the north and south, the relative species composition also differs by region. In the north, *P. vivax* is more prevalent with only a few sporadic and seasonal *P. falciparum* infections, whereas *P. falciparum* and *P. vivax* are co-endemic in the south

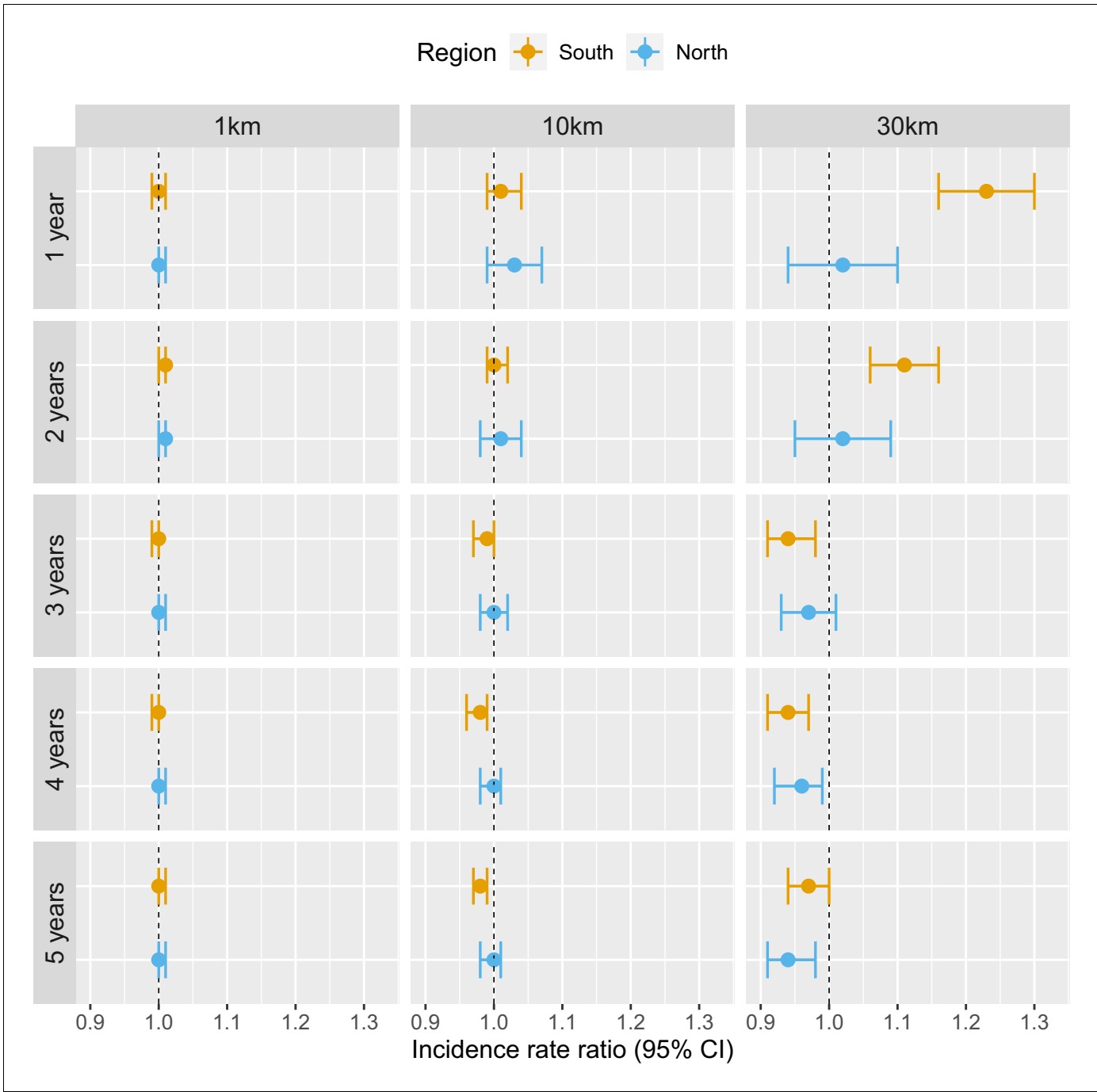

**Figure 3.** Associations between malaria incidence and a 0.1% increase in the area that experienced deforestation within 1, 10, or 30 km (left-right) of a village in the previous 1–5 years (top-down) in Lao PDR. Adjusted for the probability of seeking treatment, the spatio-temporal structure of the data, the environmental covariates selected in the model and forest cover within 30 km in the year before the deforestation temporal scale considered as well as for malaria incidence in the previous 1 and 2 years. See Materials and methods for details.

The online version of this article includes the following source data for figure 3:

**Source data 1.** Data for *Figure 3*.

(*Figure 2*). We used the co-endemicity and the larger amount of malaria case data collected in the south to assess the relationship between deforestation and malaria for both species separately.

*Table 2* and *Figure 4* show that the pattern of adjusted spatio-temporal associations identified in *Table 1* is primarily driven by *P. falciparum*, with no associations for deforestation in the near vicinity of villages (1 or 10 km) but a short-term increase (e.g. 1 year lag, IRR = 1.27, 95% CI: [1.18; 1.36]) and long-term decrease (e.g. 5-year lag, IRR = 0.83, 95% CI: [0.80; 0.87]) in *P. falciparum* malaria incidence for deforestation within 30 km of villages.

On the other hand, all the associations were attenuated for *P. vivax* infections. In the previous 2 years and within 30 km of villages, deforestation is still associated with a higher incidence of *P. vivax* (e.g. 1 year lag, IRR = 1.07, 95% CI: [1.01; 1.13]) but less so than for *P. falciparum*. However, regardless of the temporal lag or spatial scale, deforestation was no longer associated with lower *P. vivax* malaria risks.

*Appendix 1—figure 17* plots the species-specific relationships when not assuming linearity in the models.

## Alternative definitions of deforestation and interaction with forest cover

In previous models, our definition of deforestation did not distinguish between forest losses in densely forested areas and less forested areas. To explore potential interactions between deforestation and baseline forest cover, *Table 3* and *Figure 5* show how the adjusted IRR estimates vary as we consider deforestation in more densely forested pixels only (tree crown cover over 68% and 87% – see Materials and methods for rationale on thresholds). We conducted this secondary analysis only for the non-null relationships previously identified, that is, when considering a 30 km buffer radius around villages.

The associations with deforestation became more pronounced as we restricted forest losses to more forested areas: the adjusted IRR for deforestation in the previous 1 year, within 30 km of southern villages, increased from 1.16 (95% CI: [1.10; 1.22]) to 1.28 (95% CI: [1; 1.64]) when considering deforestation in areas with more than 0% and 87% tree crown cover respectively. On the other hand, the adjusted IRR for deforestation in the previous 5 years, within 30 km of southern villages, decreased from 0.94 (95% CI: [0.91; 0.97]) to 0.83 (95% CI: [0.76; 0.90]) when considering deforestation in areas with more than 0% and 87% tree crown cover, respectively. A similar trend was observed in the north, although statistical significance was not reached as frequently as in the south.

These evidence strengthen our previous results and suggest that deforestation in deep and dense forests is more closely associated with malaria incidence in villages than deforestation in less forested areas.

## Discussion

Based on a large dataset of health facility surveillance records in two regions of Lao PDR, we found evidence that deforestation around villages is associated with higher malaria incidence over the short-term but lower incidence over the long-term (e.g, in the south, within 30 km of villages: IRR = 1.16 [1.10; 1.22] for deforestation in the previous year and IRR = 0.93 [0.90; 0.97] for deforestation in the previous 3 years). Our evaluation of alternative spatial scales identified strong associations for deforestation within a 30 km radius around villages but not for deforestation in the near (10 km) and immediate (1 km) vicinity. Our results incorporated correction for the probability of seeking treatment, modeled as a function of distance to the closest health facility, as well as adjustment for several environmental covariates. Results appear driven by deforestation in densely forested areas and the patterns exhibited are clearer for infections with *P. falciparum* than for *P. vivax*.

The wide availability and longitudinal nature of malaria surveillance records collected routinely by the national program enabled exploration of the relationship between deforestation and malaria incidence over multiple spatio-temporal scales and across different levels of forest density. The spatio-temporal variability highlighted here provides insights into the causal mechanisms driving local-scale malaria incidence in the GMS. This approach not only quantified the deforestation-malaria incidence association in the GMS, but also strengthened the evidence for the key influence of forest-going populations on malaria transmission in the GMS.

This study's results echo the frontier malaria hypothesis from the Amazon region, which posits an increase in malaria incidence in the first few years following deforestation and a decrease over the

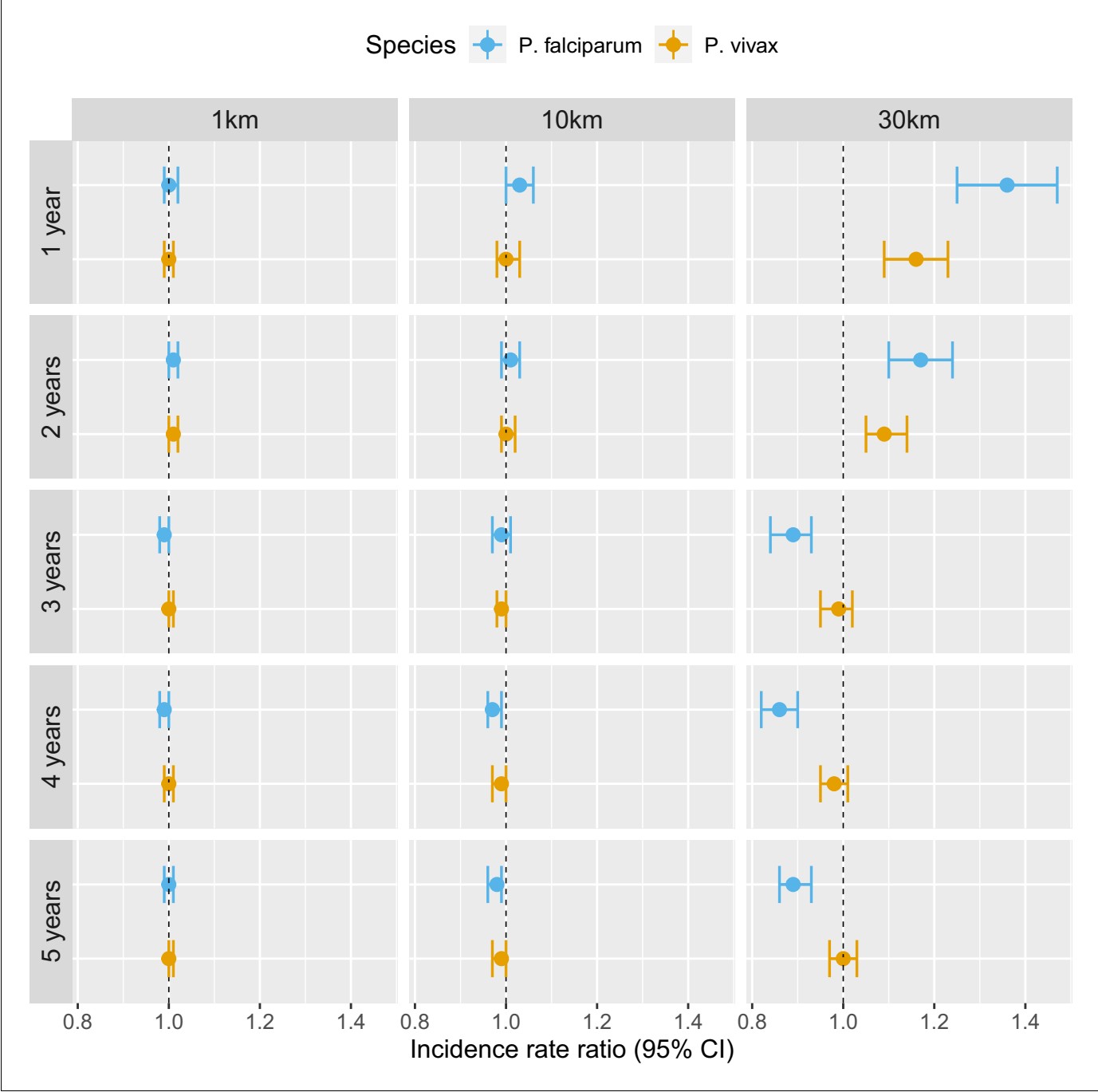

**Figure 4.** Associations between malaria incidence and a 0.1% increase in the area that experienced deforestation within 1, 10, or 30 km (left-right) of a village in the previous 1–5 years (top-down) in southern Lao PDR, differentiated by malaria species. Adjusted for the probability of seeking treatment, the spatio-temporal structure of the data, the environmental covariates selected in the model and forest cover within 30 km in the year before the deforestation temporal scale considered as well as for malaria incidence in the previous 1 and 2 years. See Materials and methods for details. The online version of this article includes the following source data for figure 4:

**Source data 1.** Data for *Figure 4*.

**Table 2.** IRR between malaria incidence and a 0.1% increase in the area that experienced deforestation within 1, 10, or 30 km (left-right) of a village in the previous 1–5 years (top-down) in southern Lao PDR, differentiated by malaria species.

Adjusted for the probability of seeking treatment, the spatio-temporal structure of the data, the environmental covariates selected in the model and forest cover within 30 km in the year before the deforestation temporal scale considered as well as for malaria incidence in the previous 1 and 2 years. See Materials and methods for details.

| Time lag | P. falciparum | | | P. vivax | | |
|---|---|---|---|---|---|---|
| | Buffer radius | | | Buffer radius | | |
| | 1 km | 10 km | 30 km | 1 km | 10 km | 30 km |
| Previous 1 year | 1 [0.99; 1.02] | 1.04 [1.01; 1.07] | 1.27 [1.18; 1.36] | 1 [0.99; 1.01] | 1 [0.97; 1.02] | 1.07 [1.01; 1.13] |
| Previous 2 years | 1 [0.99; 1.01] | 1.01 [0.99; 1.03] | 1.15 [1.08; 1.22] | 1 [0.99; 1.01] | 1 [0.98; 1.01] | 1.06 [1.01; 1.11] |
| Previous 3 years | 0.99 [0.98; 1] | 0.99 [0.97; 1.01] | 0.85 [0.80; 0.90] | 1 [0.99; 1.01] | 0.99 [0.98; 1.01] | 1.02 [0.97; 1.06] |
| Previous 4 years | 0.99 [0.98; 1] | 0.98 [0.96; 0.99] | 0.85 [0.81; 0.88] | 1 [0.99; 1] | 0.99 [0.98; 1.01] | 1.01 [0.98; 1.04] |
| Previous 5 years | 1 [0.99; 1] | 0.97 [0.95; 0.98] | 0.83 [0.80; 0.87] | 1 [1; 1.01] | 0.99 [0.98; 1] | 1.01 [0.98; 1.04] |

The online version of this article includes the following source data for Table 2:

Source data 1. Data for *Table 2*.

long term. However, we found an earlier inflexion point, 1–3 years after deforestation compared to 6–8 years in the Amazon (*de Castro et al., 2006*), most likely because of very different underlying human processes. Indeed, the frontier malaria hypothesis considers non-indigenous human settlements sprouting deeper and deeper in the forest whereas forest-going populations in the GMS are primarily members of established forest-fringe communities who regularly tour the forest overnight to hunt and collect wood (*Dysoley et al., 2008*). Industrial and agricultural projects or lucrative forest-based activities also attract mobile and migrant populations (MMPs) *Guyant et al., 2015* in remote forested areas of the GMS but not on the same scale as the politically and economically driven unique colonization of the Amazon (*de Castro et al., 2006*).

Our results are also consistent with the three previous multivariable empirical studies (*Garg, 2015*; *Pattanayak et al., 2010*; *Fornace et al., 2016*) that assessed the effect of deforestation on malaria in Southeast Asia. Our study builds on these findings by using higher resolution forest data and exploring additional spatio-temporal scales. Using biennial village census data from Indonesia between 2003 and 2008 and district-aggregated remote sensing forest data, *Garg, 2015* reported a 2–10.4% increase in the probability of a malaria outbreak in each village of districts that lost 1000 hectares of their forest cover in the same year. Using data from a 1996 cross-sectional household survey conducted in a quasi-experimental setting around a protected area in Indonesia, *Pattanayak et al., 2010* found a positive association between disturbed forest (vs undisturbed) and malaria in children under 5, again using no temporal lag. Our analysis plan was largely inspired by *Fornace et al., 2016*, which used similar high-resolution forest data (*Hansen et al., 2013*) and 2008–2012 incidence data from Sabah, Malaysia. They reported a 2.22 (95% CI: [1.53; 2.93]) increase in the *P. knowlesi* incidence rate for villages where more than 14% (<8%, being the reference) of the surrounding area (within 2 km) experienced forest loss in the previous 5 years. On the other hand, our analysis explored wider spatial scales, bypassed any coarse categorization of forest and

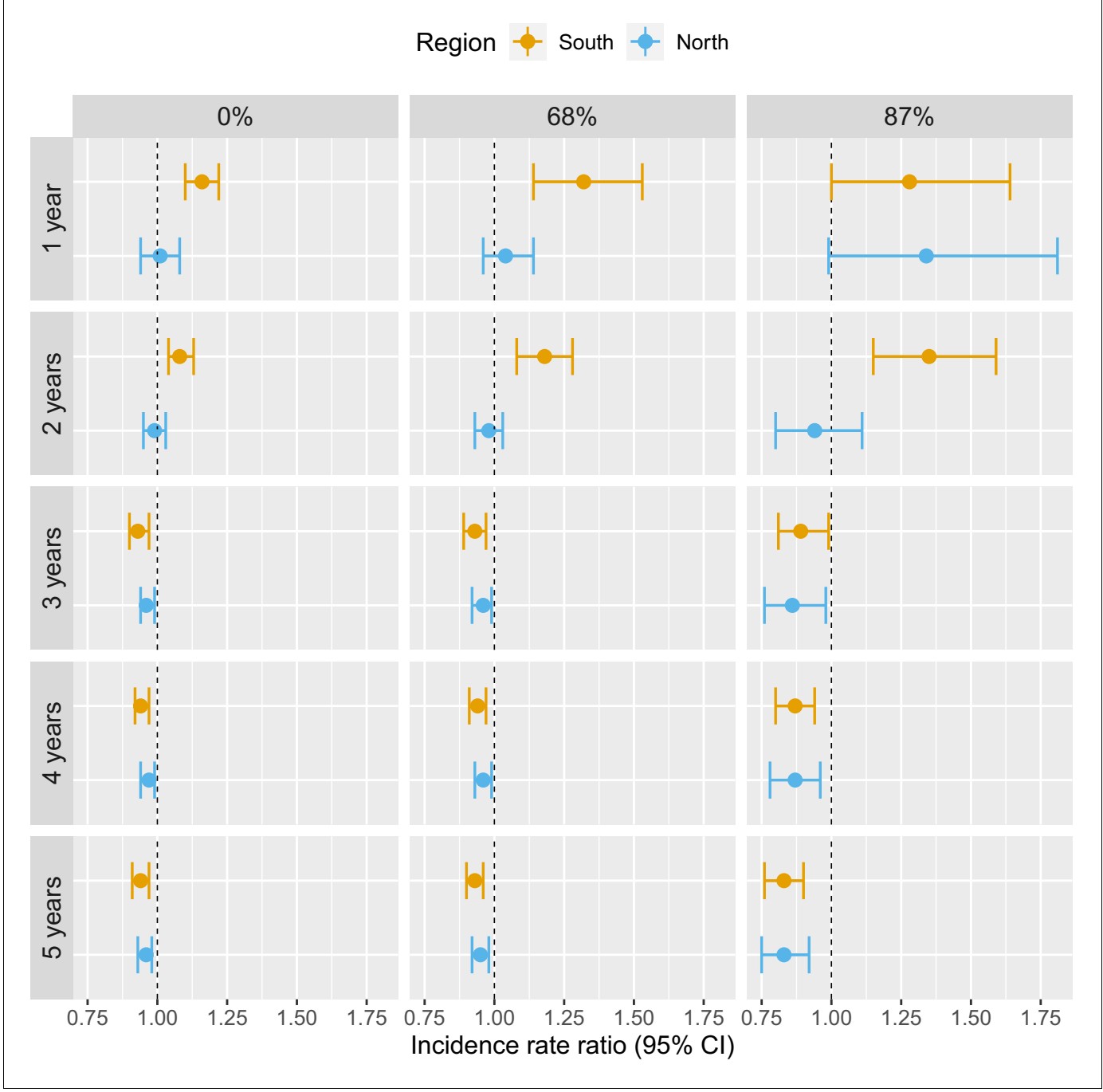

**Figure 5.** Associations between malaria incidence and a 0.1% increase in the area that experienced deforestation within 30 km of a village in the previous 1–5 years (top-down) and within areas with tree crown cover density above 0%, 68%, and 87% (left-right) in Lao PDR. Adjusted for the probability of seeking treatment, the spatio-temporal structure of the data, the environmental covariates selected in the model and forest cover within 30 km in the year before the deforestation temporal scale considered as well as for malaria incidence in the previous 1 and 2 years. See Materials and methods for details.

The online version of this article includes the following source data for figure 5:

**Source data 1.** Data for *Figure 5*.

deforestation variables, corrected incidence for treatment-seeking probability, and most importantly focused on *P. falciparum* and *P. vivax*, the dominant malaria parasites in the GMS.

**Table 3.** IRR between malaria incidence and a 0.1% increase in the area that experienced deforestation within 30 km of a village in the previous 1–5 years (top-down) and within areas with tree crown cover density above 0%, 68%, and 87% (left-right) in Lao PDR. Adjusted for the probability of seeking treatment, the spatio-temporal structure of the data, the environmental covariates selected in the model and forest cover within 30 km in the year before the deforestation temporal scale considered as well as for malaria incidence in the previous 1 and 2 years. See Materials and methods for details.

| | South | | | North | | |
|---|---|---|---|---|---|---|
| | **Deforestation within areas** | | | **Deforestation within areas** | | |
| | with tree crown cover density above | | | with tree crown cover density above | | |
| Time lag | 0% | 68% | 87% | 0% | 68% | 87% |
| Previous | 1.16 | 1.32 | 1.28 | 1.01 | 1.04 | 1.34 |
| 1 year | [1.10; 1.22] | [1.14; 1.53] | [1; 1.64] | [0.94; 1.08] | [0.96; 1.14] | [0.99; 1.81] |
| Previous | 1.08 | 1.18 | 1.35 | 0.99 | 0.98 | 0.94 |
| 2 years | [1.04; 1.13] | [1.08; 1.28] | [1.15; 1.59] | [0.95; 1.09] | [0.93; 1.03] | [0.80; 1.11] |
| Previous | 0.93 | 0.93 | 0.89 | 0.96 | 0.96 | 0.86 |
| 3 years | [0.90; 0.97] | [0.89; 0.97] | [0.81; 0.99] | [0.94; 0.99] | [0.92; 0.99] | [0.76; 0.98] |
| Previous | 0.94 | 0.94 | 0.87 | 0.97 | 0.96 | 0.87 |
| 4 years | [0.92; 0.97] | [0.91; 0.97] | [0.80; 0.94] | [0.94; 0.99] | [0.93; 0.99] | [0.78; 0.96] |
| Previous | 0.94 | 0.93 | 0.83 | 0.96 | 0.95 | 0.83 |
| 5 years | [0.91; 0.97] | [0.90; 0.96] | [0.76; 0.90] | [0.93; 0.98] | [0.92; 0.98] | [0.75; 0.92] |

The online version of this article includes the following source data for Table 3:

Source data 1. Data for *Table 3*.

Engaging in forest activities, such as logging, hunting or spending the night in the forest, has been reported as a major risk factor by many studies in the region (*Chaveepojnkamjorn and Pichai-narong, 2004*; *Das et al., 2004*; *Lansang et al., 1997*; *Erhart et al., 2005*; *Trung et al., 2004*). As countries of the GMS work toward malaria elimination, the literature stresses the key role of forest-going populations (*Guyant et al., 2015*; *Nofal et al., 2019*; *Bannister-Tyrrell et al., 2019*; *Wen et al., 2016*; *Smith and Whittaker, 2014*), although research programs highlight the challenges of accessing them (*Bennett et al., 2021* ; *Lover et al., 2019*) as well as their diversity (*Nofal et al., 2019*; *Bannister-Tyrrell et al., 2019*). To our knowledge, no previous study has leveraged geo-spatial statistical analyses to characterize the importance of forest-going populations in the GMS. Our results suggest that deforestation in dense forests (*Table 3*) around villages, particularly areas further from the village (*Table 1*), is a driver of malaria in Lao PDR. We argue that this is indicative of the existence of a key high-risk group linking the deforestation patterns identified to malaria in the villages, namely a forest-going population. Deforestation captured by remote sensing in this setting likely reflects locations and times of heightened activity in the forest areas near villages, and therefore greater human-vector contact. We suspect longer and deeper trips into the forest result in increased exposure to mosquitoes, putting forest-goers at higher risk.

We conducted this study in northern and southern Lao PDR, where the malaria species composition differs, and assessed species-specific relationships in the south where *P. falciparum* and *P. vivax* are co-endemic. Our results highlight the challenges ahead of national programs with *P. vivax* elimination after successful *P. falciparum* elimination, as increasingly mentioned in the literature (*Cotter et al., 2013*; *Kaehler et al., 2019*). This study identified a clear pattern of spatio-temporal associations between *P. falciparum* and deforestation, but these were not apparent for *P. vivax* (*Table 2*). The increase in *P. vivax* incidence in the first 2 years following deforestation was identified as well but the associations were smaller than for *P. falciparum*. Importantly, deforestation was never associated with lower risks of *P. vivax*. A recent study in the Amazon *MacDonald and Mordecai,*

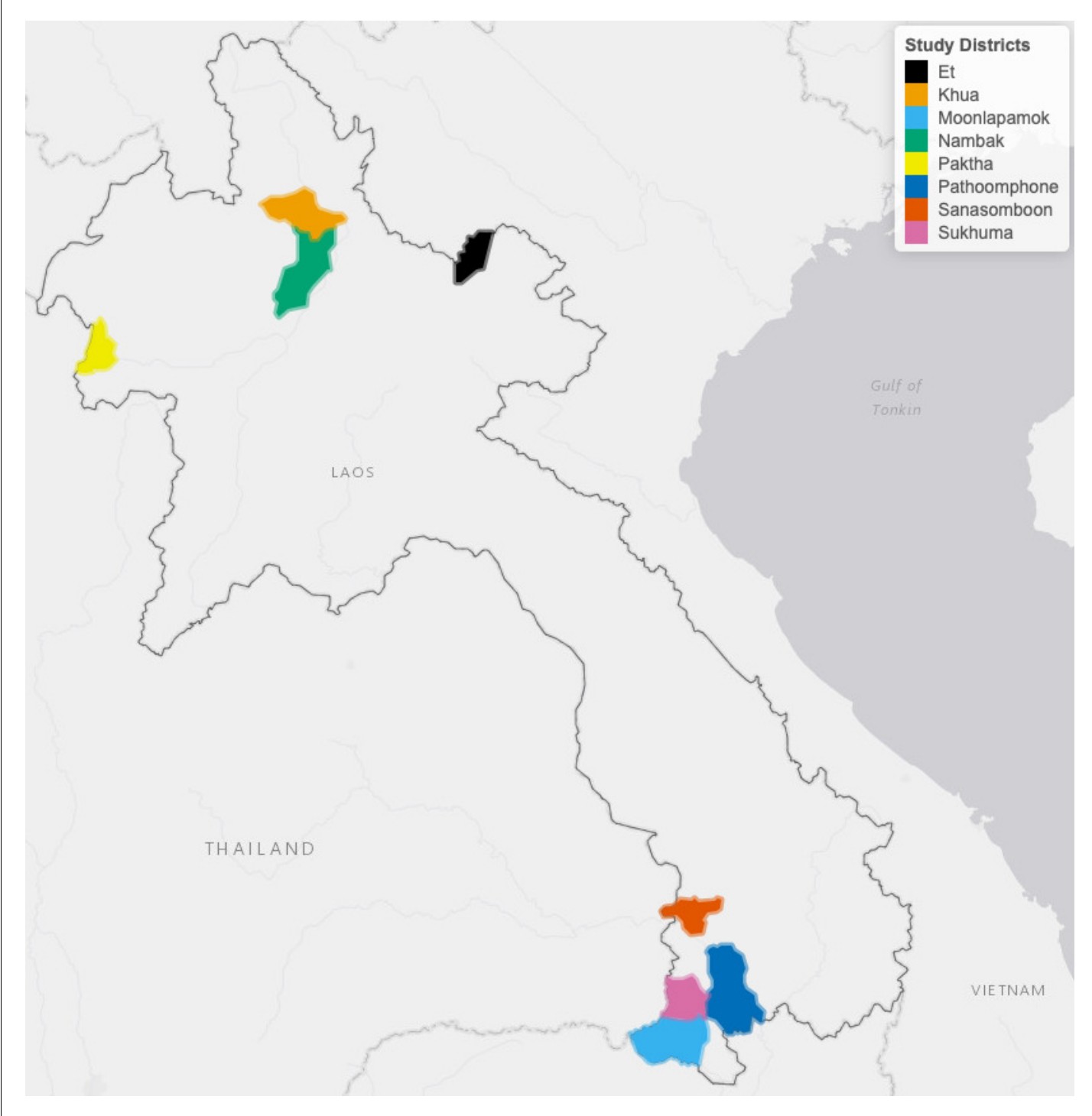

**Figure 6.** Map of study's districts.

*2019* reported a similar attenuation of the effects of deforestation on *P. vivax* compared to *P. falciparum*, most likely because of *P. vivax* parasites' ability to relapse months or even years after infection, which decouples the association between transmission and incidence data. These species-specific differences may also explain why the pattern of spatio-temporal associations between malaria and deforestation were markedly clearer in the south than in the north where *P. vivax* dominates.

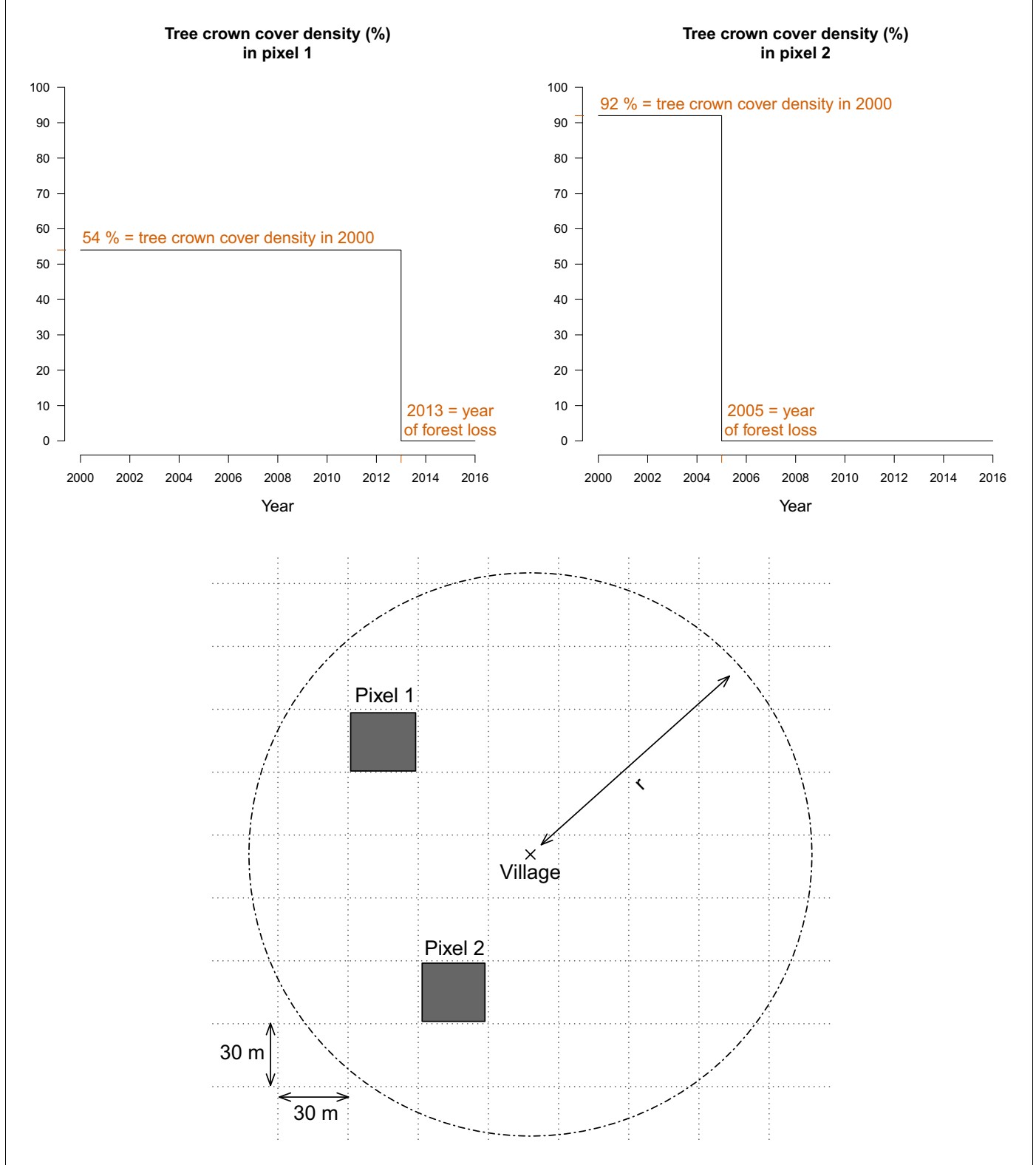

**Figure 7.** For every 30 m Landsat pixel within a buffer radius r (1, 10, and 30 km) of study's villages, the tree crown cover density in 2000 and the year of forest loss were combined to derive the deforestation and forest cover variables. The two upper plots highlight the raw data at two example pixels from the lower plot.

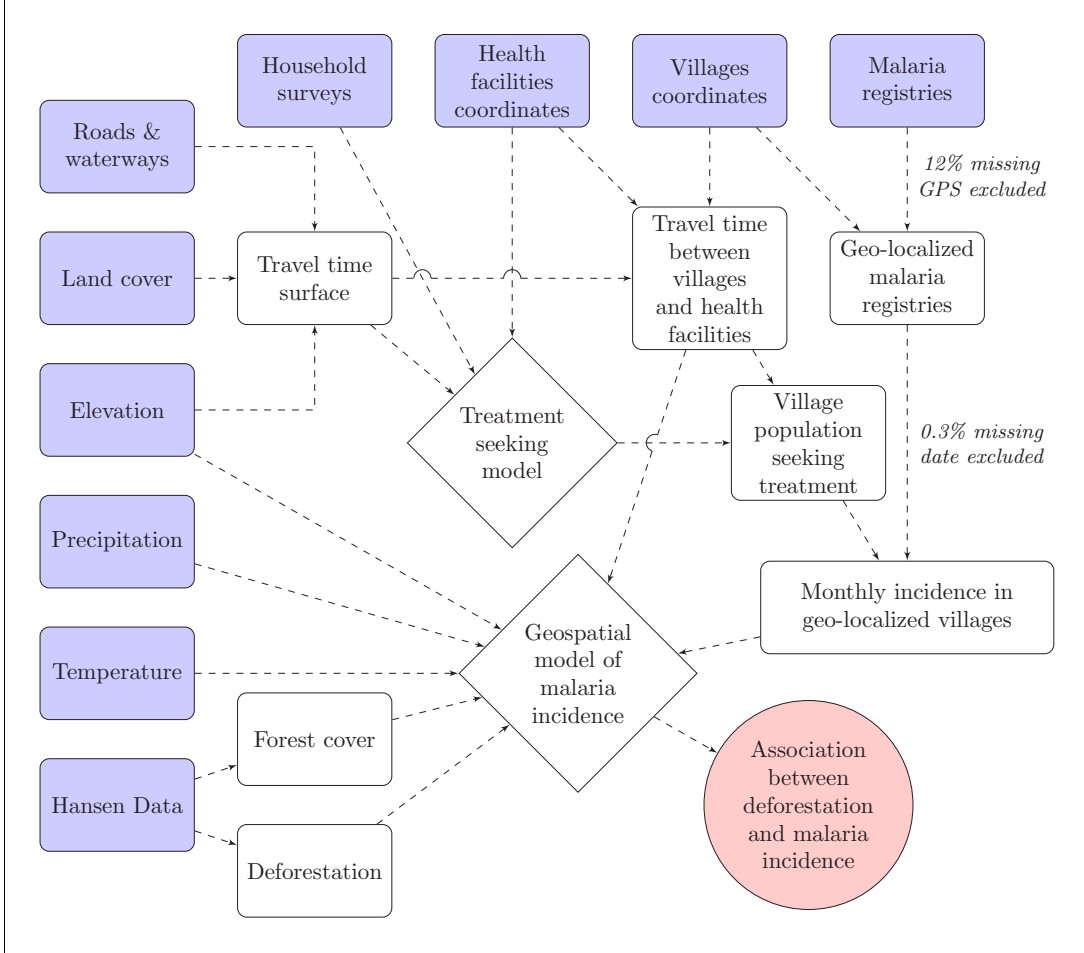

**Figure 8.** Conceptual model for our analysis showing how the raw input data (blue boxes) were combined via intermediate data (white boxes) and models (white diamonds) to produce our estimated outputs (red circle).

Our results did have some inherent limitations based upon routine health facility surveillance data. First, reliability of such records varies across and within countries of the GMS and may depend on malaria incidence level. This could lead to unmeasured residual confounding, further exacerbated by the lack of available data on malaria control activities in the region. Another challenge with these data is obtaining an accurate denominator for incidence, as not everyone attends a public health center when febrile. We addressed this issue by modeling the probability of seeking treatment as a function of travel time to the closest health facility using data from two cross-sectional surveys. Last, the village-level geo-referencing of malaria registries ignores the possibility that patients may become infected elsewhere. Unfortunately, these surveillance records did not include information about patients' forest-going trips. Research to track and analyse micro-scale movements of forest-goers is needed to understand how they interact with the forest and where are the foci of infection.

The forest data we used has also been criticized, in particular for not distinguishing tropical forests from agroforestry (*Tropek et al., 2014*; *Hansen et al., 2014*) or man-made from natural causes of deforestation. The lack of temporal resolution for the forest gain variable (2000–2017 aggregate) as well as the assumption that forest loss happens all in 1 year are additional limitations of these data. Finally, our relative measure of deforestation, key to consistently compare the effects across different spatial scales, also implies that a 0.1% of the area that experienced forest loss within 30 km of a village is a much larger area (~280 hectares) than within 1 km (~0.3 hectare) and should be interpreted cautiously.

In conclusion, this study assessed the relationship between deforestation and malaria in Lao PDR. Our approach leveraged surveillance records collected by the national malaria program and high-

resolution forest data and rigorously explored the spatio-temporal pattern of associations. As countries of the GMS work toward malaria elimination, our results highlight the challenges to transition from *P. falciparum* to *P. vivax* elimination, confirm and characterize the importance of high-risk populations engaging in forest activities and suggest malaria programs may benefit from monitoring areas of on-going deforestation using remotely sensed data.

## Materials and methods

### Study site and population

Lao PDR has seen a 92% reduction in cases between 2000 (280,000) and 2010 (23,000) (*Lao PDR, 2016*). Much of this progress has been attributed to heightened funding and better testing and treatments (*Okayas, 2018*).

This study was conducted in eight districts (*Figure 6*) to leverage the ecological and epidemiological diversity of Lao PDR. Four districts (Moonlapamok, Pathoomphone, Sanasomboon, and Sukhuma) are situated in the southern province of Champasak where both *P. falciparum* and *P. vivax* are endemic. The four other districts (Et, Paktha, Nambak, and Khua) each come from one of four northern provinces (Bokeo, Huaphanh, Phongsaly, Luang-Prabhang) where *P. vivax* is endemic but *P. falciparum* has reached historical lows (*Lao national malaria database (dhis2), 2018*).

The four districts in the north were chosen in consultation with district and provincial level malaria staff to represent the epidemiology of malaria in the region. They were selected as part of a cross-sectional survey designed to assess the prevalence and risk factors for malaria in northern Lao PDR (*Lover et al., 2018*). This region is very mountainous and characterized by a diverse climate, low-population density and limited road access (*UNFPA, 2016*). Land clearing using fires for agriculture is customary.

The four districts in the south were selected within a larger cluster randomized controlled trial (RCT) study designed to assess the effectiveness of high-risk group targeted active case detection in southern Lao PDR (*Lover et al., 2019*), where more than 95% of the country malaria burden is concentrated (*Lao national malaria database (dhis2), 2018*). This region is characterized by a moderately hilly and forested terrain and a workforce primarily engaged in forest-based and agricultural activities (*Bennett et al., 2021*).

When designing the study, in collaboration with the national control program, we purposefully excluded regions where we knew large programmatic activities where being implemented.

### Malaria data

#### Malaria case data

We conducted a retrospective review of malaria registries recorded at all health centers in the study districts between January 2013 and December 2016 in the north and between October 2013 and October 2016 in the south. The registries included information on every patient that was tested (RDT and/or microscopy) for malaria at the health center. Date, species-specific test results, demographic variables (age, gender, and occupation) and the village of residence of the patient were recorded in the registries. With help from local Lao experts, village names were matched to a geo-registry of all villages in Lao PDR compiled from the 2005 and 2015 national census *Lao national census, 2020* and provided by the Center for Malariology, Parasitology and Entomology (CMPE). The geo-registry contains GPS coordinates and population of Lao PDR's villages. Unmatched records and records with missing date were removed from the analysis. Finally, these data were aggregated to extract the monthly village-level malaria incidence.

#### Treatment-seeking data

One issue with using passive surveillance data is that not everyone will seek treatment at a public health facility for a febrile illness, which can lead to an underestimate of the true incidence, if not accounted for. To correct for that, we modeled the probability that an individual in a given village of the study's district would seek treatment at a public health facility when febrile. We assumed that such probability is essentially driven by the travel time to the closest health facility. See Appendix 1 - S1 for methods used to calculate travel times to closest health facilities.

To model the probability of seeking treatment, we used data from two cross-sectional household surveys conducted in the eight districts where registries were collected. In the north, 1480 households across 100 villages were surveyed in September-October 2016 (*Lover et al., 2018*). In the south, 1230 households across 56 villages were surveyed in the baseline assessment of the RCT (*Lover et al., 2019*) in December 2017. In particular, survey respondents were asked whether or not they would seek treatment at the closest health facility for a febrile illness and GPS coordinates of their household were recorded.

We then used the cross-sectional surveys to model the probability of seeking treatment (at a public health facility, implicit from now on), $\theta$, as a function of travel time to the closest health facility, $\tau$ (*Equation 1*). To account for the correlation structure induced by the stratified sampling approach used in the surveys, we modeled the number of successes (febrile patients seeking treatment), $S_{h,v}$, at the household level and included a random intercept for village in the logistic regression.

$$S_{h,v} \sim Bin(\theta_{h,v}, N_{h,v})$$
$$\mathrm{logit}(\theta_{h,v}) = \alpha_0 + \alpha_1 * \tau_{h,v} + \alpha_v$$

(1)

where $N_{h,v}$ is the number of febrile individuals in household h of village v and $\alpha_v \sim \mathcal{N}(0, \sigma_\alpha)$.

We fit the models separately in the north and in the south and used the region-specific model to predict the probability of seeking treatment at all villages of the study districts based on their distance to the closest health facility. The population who seek treatment was then calculated by multiplying the village population by the probability of seeking treatment. See Appendix 1 - S2 for travel times and treatment-seeking probabilities results.

## Forest data

For every 30 m pixel in Lao PDR, tree crown cover density for the year 2000 and year of forest loss between 2000 and 2017, were obtained from *Hansen et al., 2013*. These layers were produced using decision tree classifiers on Landsat remote sensing imagery (*Hansen et al., 2013*). Trees are defined as 'all vegetation taller than 5 m in height' (*Hansen et al., 2013*) and forest loss as 'the removal or mortality of all tree cover in a Landsat pixel' (*Hansen et al., 2014*). For example, as depicted in *Figure 7*, the Hansen data indicates that the tree crown cover in 2000 in pixel 1 is 54%, meaning that 54% of the 30 m pixel is covered by vegetation taller than 5 m. The Hansen data also indicates that forest loss occurred in pixel in 2013, meaning that all of the tree canopy disappeared in 2013.

### Deforestation variable

To define our primary exposure variable, for all villages in the study districts and year of the study period, we calculated the percent area within a buffer radius of 1, 10, and 30 km that experienced forest loss in the previous 1, 2, 3, 4, and 5 years (*Figure 7*). These distances were chosen to explore a range of spatial scales at which the forest environment may be differentially relevant for village-based populations and forest-goers. To explore potential interactions between deforestation and forest cover, we computed an alternate exposure variable, restricting to areas that both experienced forest loss and had a tree crown cover density above 68% and 87%. Those thresholds are limits of the inter-quartile range (IQR) of the distribution of tree crown cover density in any 30 m pixels within 10 km of study's villages that experienced forest loss between 2000 and 2017. This alternate definition captures deforestation activities occurring in areas with denser forest cover.

### Forest cover variable

We also combined the two Hansen layers to produce annual tree crown cover maps of the study districts, assuming no changes prior to the year of forest loss but setting to 0 the pixel tree crown cover density afterwards (*Figure 7*). For all villages in the study districts and year of the study period, we calculated the average tree crown cover density within a buffer radius of 1, 10, and 30 km and for 0, 1, 2, and 3 year lags. This is a secondary exposure, adjusted for in the primary analysis.

## Environmental covariates

Village population sizes were needed to estimate monthly malaria incidence. 2005 and 2015 population estimates for the 491 villages of study districts were obtained from the national census

*Lao national census, 2020*. The annual population growth rate (3.7%) was used to impute population values for two villages missing 2005 estimates and for two villages missing 2015 estimates. Then, village-level population growth rates were used to estimate villages' population per year between 2008 and 2016, assuming linear annual growth rate (median = 1.7%, IQR = [0%; 4.5%]).

Altitude, temperature, rainfall, and access to health care were considered as potential village-level confounders of the relationship between malaria and forest cover factors. Travel time to closest health facility, computed for the treatment-seeking model, was used as a proxy for health care access and villages' remoteness. Altitude was extracted from SRTM (*Jarvis et al., 2008*) 1 km resolution layers. Monthly average day and night temperature were extracted from MODIS 1 km resolution product (MOD11C3 *Wan et al., 2015*). Finally, monthly total rainfall was extracted from CHIRPS (*Funk et al., 2014*) 1 km resolution publicly available data. The average and standard deviation of the annual total precipitation and the average monthly temperature from the monthly time series was computed over the 2008–2012 period, which corresponds to the 5 year time period directly before our malaria data (2013–2016). This 'long-term' aggregation of the climatic variables is included in the model to capture the spatial differences in overall climate between the villages of our study area. To account for the seasonal effect of these climatic variables, monthly temperatures and precipitation in the previous 1, 2, and 3 months were also extracted, as well as the average temperatures and total precipitation over the previous 1, 2, and 3 months (seven 'short-term' variations: in current month, in previous 1, 2, or 3 months and aggregated over current and previous 1, 2, or 3 months). See 'Details on covariates' below.

Altitude was missing for one village and we used an online elevation finder tool (FreeMapTools) for imputation. Temperature was missing for 2.4% of the village-months over the study period, most likely because of cloud coverage of the MODIS imagery. Monthly temperature was never missing more than 2 years in a row at villages of the study's districts and we imputed the temperature of the same month of the following year (or prior year when needed), adjusting for average district-level monthly temperature differences between the 2 consecutive years. Monthly rainfall was not missing at any of the villages.

## Statistical analysis
### Statistical model
To model malaria incidence (*Equation 2*), the number of positive cases $Y_{v,t}$ at village v over month t was modeled using a generalized additive model (GAM) (*Wood, 2017*). To account for overdispersion, a negative binomial distribution was used, including an additional variance parameter $\nu$. The probability of seeking treatment $\theta_v$, estimated from the treatment-seeking model, was multiplied by the village population $Pop_{v,t}$ to derive the population seeking treatment, $Pop_{v,t}^{seek}$. This was included as an offset term in the incidence model. Spatial autocorrelation was accounted for by the bivariate thin plate spline smoothing function on coordinates, $f(Lat, Long)$ and village random intercepts were included. A non-linear temporal trend was also included with the smoothing function on month, $f(t)$. Finally, the primary exposure, deforestation, and potential environmental confounders, including forest cover, were modeled with splines in $f(X_{v,t}^i)$. Splines add up polynomial basis functions in between knots and allow to control for very flexible relationships with covariates and spatio-temporal trends. Regularization was used to integrate model selection into the model fitting step by adding an extra penalty to each term so that the coefficients for covariates can be penalized to zero, also meaning that splines can be kept minimal if the data does not support more flexibility. See *Figure 8* for a graphical visualization of our conceptual model for this analysis.

$$Y_{v,t} \sim \mathrm{NegBin}(E[Y_{v,t}], \nu)$$
$$\log(E[Y_{v,t}]) = \log(\mu_{v,t} * Pop_{v,t}^{seek}) = \log(\mu_{v,t}) + \log(Pop_{v,t} * \theta_v)$$
$$\log(\mu_{v,t}) = \sum_i \beta^i * f(X_{v,t}^i) + f(t) + f(Lat, Long) + \beta_v$$

(2)

with $\beta_v \sim \mathcal{N}(0, \sigma_\beta)$.

We ran 15 models separately in the north and the south, each varying the buffer radius (1, 10, and 30 km) and temporal scale for deforestation (previous 1, 2, 3, 4, and 5 years). The coefficients of the linear effect for deforestation were extracted and exponentiated to get the incidence rate ratio

(IRR) associated with a 0.1% increase in the percent area that experienced forest loss around villages.

## Secondary analyses

As secondary analyses, we ran the same models, separately by malaria species (*P. falciparum* and *P. vivax*), leveraging the large amount of data and co-endemicity in the south. We also used our alternative definitions for deforestation, restricted to areas that both experienced forest loss and had a tree crown cover density above 68% and 87%, to explore the interaction between deforestation and the amount of forest cover.

To further strengthen the robustness of our analysis, we conducted a sensitivity analysis where villages' populations in the surveillance system registries were not adjusted for the probability of seeking treatment. See Appendix 1 - S3.

## Details on covariates

To prevent collinearity in the final model, for each of the three monthly climatic variables (precipitation, day and night temperature), we first selected the one of its seven 'short-term' variations (in current month, in previous 1, 2, or 3 months and aggregated over current and previous 1, 2, or 3 months) that provided the best AIC fit in an univariate model, solely adjusted for the spatio-temporal structure of the data ($f(t)$, $f(Lat, Long)$ and village random intercepts). See Appendix 1 - S6.

Malaria incidence in the previous 1 and 2 months were included in the model. Results in Appendix 1 - S4 show this was necessary to fully address temporal autocorrelation and led to a better AIC fit. Different shape of the temporal trend *f(t)* were also explored (up to 25 spline knots, auto-regressive, cyclic cubic spline) but none accounted for temporal autocorrelation better.

In a preliminary analysis, before including the deforestation variable, we ran our model in ***Equation 2*** with forest cover as the primary exposure. We ran 12 models separately in the north and the south, each varying the buffer radius (1, 10, and 30 km) and temporal scale (0, 1, 2, and 3 year lag) for the forest cover variable. The coefficients of the linear effect for forest cover were extracted and exponentiated to get the incidence rate ratio (IRR) associated with a 1% increase in the average tree crown cover density around villages (***Appendix 1—table 3***). The model including average tree crown cover density within 30 km of villages with no temporal lag provided the smallest AIC value. In the final models with the deforestation variables we therefore included the average tree crown cover density within 30 km of villages in the starting year of the temporal scale for the deforestation variable considered (e.g. 3-year lag in the model with percent area that experienced forest loss in previous 3 years as the deforestation variable) to adjust for baseline forest cover.

## Acknowledgements

For their expertise and assistance, we thank Michelle Roh (Study design and interpretation), Ricardo Andrade Pacheco (GAM and spatial modeling), Alemayehu Midekisa (Remote sensing), Stephen Shiboski (Statistical analyses) and Maria Glymour (Causal inference and manuscript writing). FR was funded was the Bill and Melinda Gates Foundation (Grant ID OPP1116450).

## Additional information

### Funding

| Funder | Grant reference number | Author |
| --- | --- | --- |
| Bill and Melinda Gates Foundation | OPP1116450 | Francois Rerolle<br>Emily Dantzer<br>Andrew A Lover<br>Bouasy Hongvanthong<br>Hugh JW Sturrock<br>Adam Bennett |

The funders had no role in study design, data collection and interpretation, or the decision to submit the work for publication.

## Author contributions

Francois Rerolle, Conceptualization, Data curation, Formal analysis, Investigation, Visualization, Methodology, Writing - original draft, Writing - review and editing; Emily Dantzer, Data curation, Project administration, Writing - review and editing; Andrew A Lover, Data curation, Funding acquisition, Project administration, Writing - review and editing; John M Marshall, Hugh JW Sturrock, Supervision, Methodology, Writing - review and editing; Bouasy Hongvanthong, Project administration, Writing - review and editing; Adam Bennett, Conceptualization, Supervision, Funding acquisition, Methodology, Project administration, Writing - review and editing

## Author ORCIDs

Francois Rerolle https://orcid.org/0000-0002-3837-5700
Andrew A Lover http://orcid.org/0000-0002-2181-3559
John M Marshall http://orcid.org/0000-0003-0603-7341

## Ethics

Human subjects: This study was approved by the National Ethics Committee for Health Research at the Lao Ministry of Health (Approval #2016-014; 8/22/2016) and by the UCSF ethical review board (Approvals #16-19649 and #17-22577). The informed consent process was consistent with local norms, and all study areas had a consultation meeting with, and approvals from, village elders. All participants provided informed written consent; caregivers provided consent for all children under 18, and all children aged 10 and above also provided consent directly. The study was conducted according to the ethical principles of the Declaration of Helsinki of October 2002.

## Decision letter and Author response

Decision letter https://doi.org/10.7554/eLife.56974.sa1
Author response https://doi.org/10.7554/eLife.56974.sa2

## Additional files

### Supplementary files

- Source code 1. Code for *Figure 2*.
- Source code 2. Code for *Figure 3*.
- Source code 3. Code for *Figure 4*.
- Source code 4. Code for *Figure 5*.
- Transparent reporting form

### Data availability

All data generated or analysed during this study are included in the manuscript and supporting files. Source data files have been provided for Figures 2, 3, 4 and 5 and for Tables 1, 2 and 3.

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

## Appendix 1

### S1: Travel times methods

To calculate the travel time along a path linking any two points of the map, we defined a transition matrix that gives the speed at which one may travel between two adjacents pixels. We followed the parameterization suggested by *Alegana et al., 2012* and demonstrated by *Sturrock et al., 2014* (see *Appendix 1—table 1*), which first uses Toblers' hiking function to specify the travel speed between two points of different altitudes. Intuitively, it is faster to travel downhill than uphill. Second, the speed is adjusted based on the type of landcover traveled through: a forested or a flooded area for instance slows you down. Last, the network of roads and major rivers may be used to catch a bus or a boat and therefore increases the travel speed. Altitude (SRTM 90 m, *Jarvis et al., 2008*) was aggregated and resampled at the land cover (ESA GlobCover 2009 Project *European space agency, 2009*) 300m-resolution and roads and waterways from Open Street Map (*Open street map, 2020*) were rasterized to calculate the transition matrix all across Lao PDR. The 'raster' package in R (*R Development Core Team, 2020*) was used.

**Appendix 1—table 1.** Data used to parameterized the transition matrix with the travel speed between any two adjacent pixels of the map.

| Data layer | Category | Speed (km/h) |
|---|---|---|
| Digital elevation (slope) | 0°(flat) | 5 |
| | 5°(uphill) | 3.71 |
| | −5°(downhill) | 5.27 |
| Land cover | Cropland | No adjustment |
| | Artificial and bare areas | No adjustment |
| | Open deciduous forest | 0.8 * Hiking speed |
| | Sparse herbaceous | 0.8 * Hiking speed |
| | Closed deciduous forest | 0.6 * Hiking speed |
| | Herbaceous | 0.6 * Hiking speed |
| | Flooded | 0.5 * Hiking speed |
| | Other forest cover | 0.4 * Hiking speed |
| | Water | 0.2 * Hiking speed |
| Roads and rivers | Motorway/trunk | 80 |
| | Primary/secondary | 60 |
| | Tertiary/unclassified | 10 |
| | Major rivers | 5 |

We then used the Djisktra's algorithm from the R-package igraph (*Csardi and Nepusz, 2006*) and the gdistance package (*van Etten, 2017*) to find the fastest route between every village (or every household in the cross-sectional surveys) and its closest health facility. Coordinates of health facilities across Lao PDR came from the 2017 stratification exercise and were provided by CMPE. We authorized travel through non-study districts but not across international borders.

### S2: Travel times and treatment-seeking results

*Appendix 1—figure 1a* shows how the travel time to closest health facility varies across Champasak province in southern Lao PDR, influenced by both distance and road connectivity. *Appendix 1—figure 1c* presents a right-skewed distribution of travel time from study villages to the closest health facility. Most villages are within 2 hr of the closest health facility but some are as far as 6 hr away. The distribution is similar for villages in the northern and southern study districts.

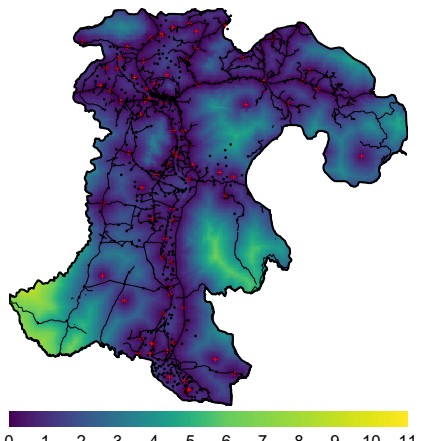

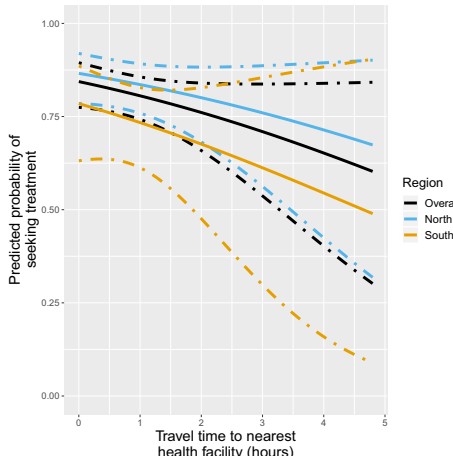

(a) Travel time (hours) to closest health facility (red crosses) in Champasak province, southern Lao PDR. Black dots represent villages and lines show main roads that may be used to travel.

(b) Modeled relationship between treatment-seeking probability and travel time to closest health facility. Dashed lines represent the 95% confidence boundaries.

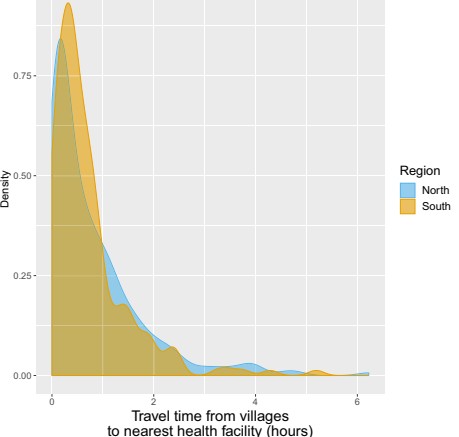

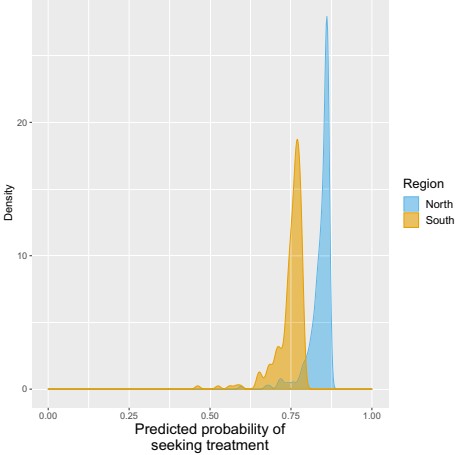

(c) Distribution of travel time (in hours) from villages to closest health facilities.

(d) Distribution of the predicted probability of seeking treatment.

**Appendix 1—figure 1.** Treatment-seeking modeling plots. Note that treatment-seeking at public health facilities is implied all along the manuscript.

In the southern household survey, 243 individuals reported fever in the past 2 weeks. 225 (92.6%) of them, from 156 households, answered whether or not they sought treatment and were included in the treatment-seeking model. 219 (97.3%) reported seeking treatment and they all reported where they did so: 154 (70.3%) of them sought treatment at a public health facility (Village malaria worker (VMW), health center, district hospital or provincial hospital) and would therefore appear in the malaria registries collected.

In the northern household survey, 378 individuals reported fever in the past 2 weeks. 360 (95.2%) of them, from 297 households, answered whether or not they sought treatment and were included in the treatment-seeking model. A total of 283 (78.6%) reported seeking treatment. Only 40 (14.1%) of them reported where they did so but all of them sought treatment at a public health facility and we therefore upweighted the population that sought treatment at a public health facility accordingly.

Most surveyed households included in the treatment-seeking model were within 2 hr of travel time to the closest health facility but some were almost 5 hr away (*Appendix 1—figure 12*). *Appendix 1—figure 1b* shows the modeled relationship between the probability of seeking treatment (at a public health facility, implied from now on) and distance to the closest health facility. For villages within the same 300 m$^2$ pixel as a health facility (estimated travel time of 0 hr), the predicted probability of seeking treatment was 0.87 (95% CI: [0.79; 0.92]) in the north and 0.78 (95% CI: [0.63; 0.89])

in the south. A 1 hr increase in travel time to the closest health facility was associated with a similar 0.79 (95% CI: [0.55; 1.13]) reduction in the odds of seeking treatment in the north and 0.76 (95% CI: [0.43; 1.34]) in the south, almost reaching statistical significance when pooling data from both regions: 0.77 (95% CI: [0.56; 1.04]). *Appendix 1—figure 1d* shows the resulting distribution for the probability of seeking treatment for all villages in study's districts. Monthly village-level malaria incidence was adjusted accordingly.

## S3: Sensitivity analysis

We conducted a sensitivity analysis where village population at risk of appearing in the surveillance system registries were not adjusted for the probability of seeking treatment. The effect estimates and confidence intervals were virtually unchanged, strengthening the robustness of our primary analysis (*Appendix 1—table 2* below).

**Appendix 1—table 2.** IRR associated with a 0.1% increase in forest loss.
Adjusted for the spatio-temporal structure of the data, the environmental covariates selected in the model and forest cover within 30 km in the year before the deforestation temporal scale considered and malaria incidence in the previous 1 and 2 months. See Materials and methods for details. Sensitivity analysis: village population unadjusted for probability of seeking treatment.

| Time lag | South | | | North | | |
|---|---|---|---|---|---|---|
| | Buffer radius | | | Buffer radius | | |
| | 1 km | 10 km | 30 km | 1 km | 10 km | 30 km |
| Previous 1 year | 1 [0.99; 1.01] | 1.01 [0.99; 1.04] | 1.16 [1.10; 1.22] | 1 [1; 1.01] | 1.03 [1; 1.07] | 1.01 [0.94; 1.08] |
| Previous 2 years | 1 [0.99; 1.01] | 1 [0.98; 1.01] | 1.09 [1.04; 1.13] | 1 [1; 1.01] | 1.01 [0.99; 1.04] | 0.98 [0.94; 1.01] |
| Previous 3 years | 0.99 [0.99; 1] | 0.98 [0.97; 1] | 0.93 [0.90; 0.97] | 1 [1; 1.01] | 1.01 [0.99; 1.02] | 0.96 [0.93; 0.99] |
| Previous 4 years | 0.99 [0.99; 1] | 0.98 [0.97; 0.99] | 0.94 [0.92; 0.97] | 1 [1; 1.01] | 1 [0.99; 1.02] | 0.97 [0.94; 0.99] |
| Previous 5 years | 1 [0.99; 1] | 0.97 [0.96; 0.99] | 0.94 [0.91; 0.97] | 1 [1; 1.01] | 1.01 [0.99; 1.02] | 0.95 [0.93; 0.98] |

## S4: Inclusion of malaria cases in previous months

*Appendix 1—figure 2* shows residual temporal auto-correlation plots in models from equation 2, when malaria incidence in previous 1 and 2 months are included or not. These plots show that including covariates for malaria incidence in the previous 1 and 2 months is necessary to address residual temporal autocorrelation and keep each lag-wise individual autocorrelation estimate below 5%. The plots presented here are for the model in the south with a 30 km buffer radius and a 1-year temporal lag but similar results were observed across all 15 models both in the north and in the south.

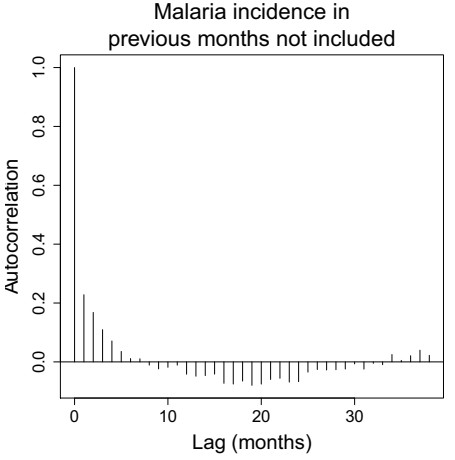

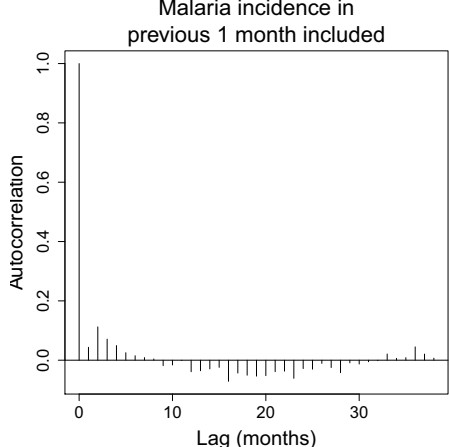

(a) Residual temporal autocorrelation when malaria incidence in previous months is not included.

(b) Residual temporal autocorrelation when malaria incidence in previous 1 month is included but not in previous 2 months.

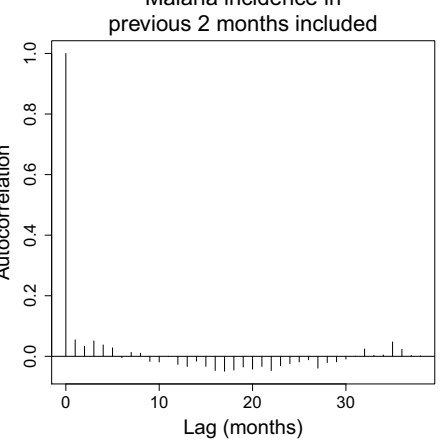

(c) Residual temporal autocorrelation when malaria incidence in previous 1 and 2 months are included.

**Appendix 1—figure 2.** Residual temporal autocorrelation when malaria incidence in previous 1 and 2 months are included or not.

The AIC fit also substantially improved from 18,667 when no malaria incidence is included to 18,152 when malaria incidence in previous month is included and to 17687 when both malaria in the previous 1 and 2 months are included.

## S5: Additional results

### Environmental covariates

*Appendix 1—figure 3* shows the relationship – via their individual contribution $\beta * f(X)$ in equation 2 – between malaria incidence and the environmental covariates included in the model (30 km radius and 1-year temporal lag). Thses plots show that relationships differ slightly by region although the range covered by the environmental variables also differs by region. We also see the effect of regularization, that penalized some covariates to zero, like our long-term precipitation covariates. This penalization happened more frequently in the north, where we had much less data. Note that 95% confidence intervals (see *Appendix 1—figure 13*) have been hidden for better visualization. The larger amount of data in the south also allowed the identification of more precise relationships than in the north.

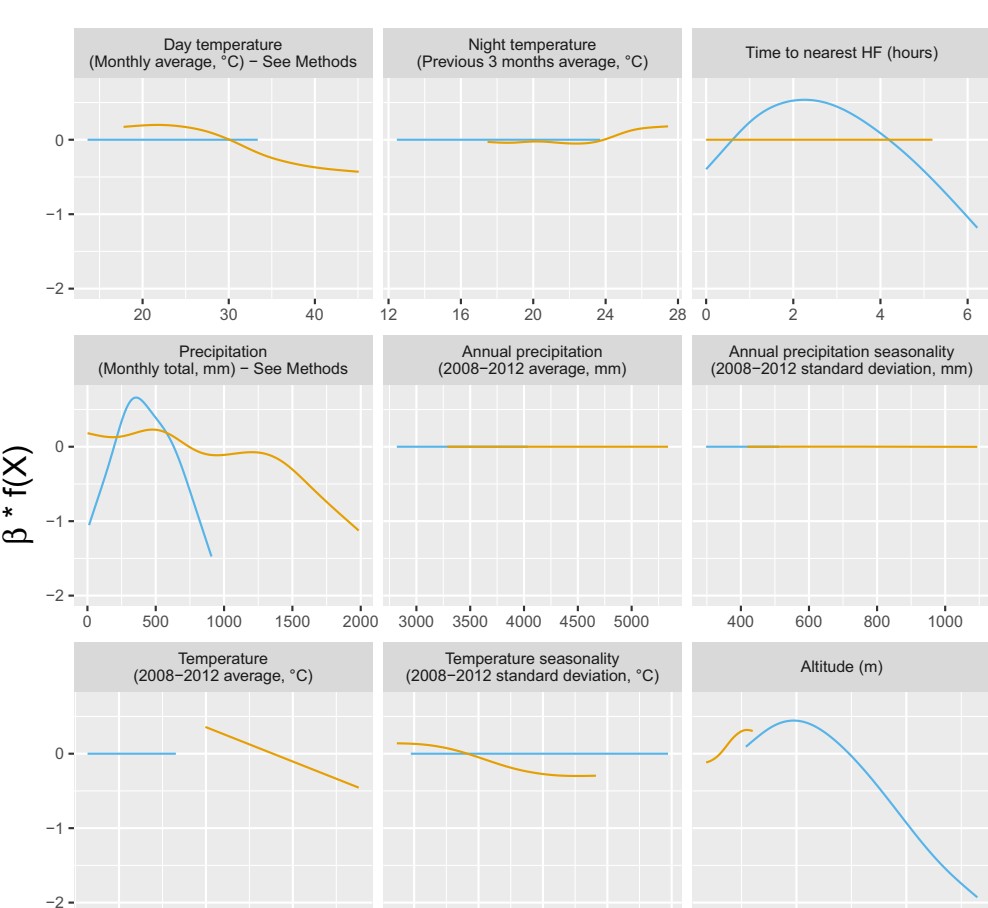

**Appendix 1—figure 3.** Relationships between malaria incidence and the environmental covariates in the multivariable model described in equation 2 (30 km radius and 1-year temporal lag), additionally adjusted for the probability of seeking treatment, the spatio-temporal structure of the data ($f(t)$, $f(Lat, Long)$ and village random intercepts) and malaria incidence in the previous 1 and 2 months. See Materials and methods for details. Note that 95% confidence intervals (see *Appendix 1—figure 13*) have been hidden for better visualization.

## Temporal trend

*Appendix 1—figure 4* shows the relationship – via its individual contribution $\beta * f(t)$ in equation 2 – between malaria incidence and the temporal trend included in the model (30 km radius and 1-year temporal lag). These plots show that relationships are quite similar in both regions with an increase in 2014, followed by a plateau in 2015 and a decrease in 2016. The larger amount of data in the south also allowed the identification of a more precise relationship than in the north.

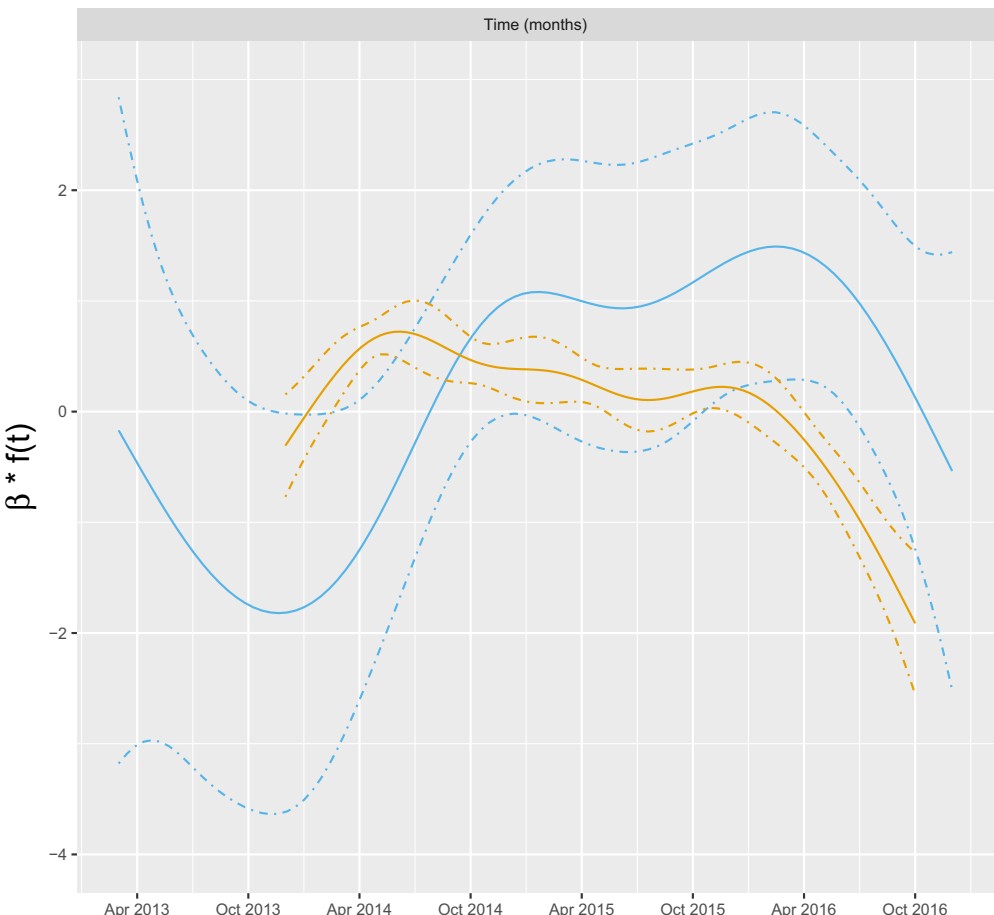

**Appendix 1—figure 4.** Relationships between malaria incidence and the temporal trend in the multi-variable model described in equation 2 (30 km radius and - year temporal lag), additionally adjusted for the probability of seeking treatment, the spatial structure of the data ($f(Lat, Long)$ and village random intercepts) and malaria incidence in the previous 1 and 2 months. See Materials and methods for details.

## Forest cover

*Appendix 1—table 3* shows the incidence rate ratio (IRR) associated with forest cover, measured by a 1% increase in the average tree crown density, in current and previous 3 years within 1, 10, and 30 km of villages. Forest cover within 1 km of a village was not associated with malaria incidence rate in either the south or the north, regardless of the temporal lag. However within 10 and 30 km of a village, increased forest cover tended to be associated with higher malaria incidence rates both in the north and the south (e.g. 30 km buffer, 1-year lag, IRR = 1.09, 95% CI: [1.03; 1.15] in the south; IRR = 1.12, 95% CI: [0.99; 1.26] in the north). The associations were higher when considering a larger spatial scale (30 km) but were already statistically significant for a 10 km buffer radius in the south. None of the associations reached statistical significance in the north, where the sample size is small. The temporal scale considered did not affect the associations much. Statistical significance wasn't necessarily reached for all the associations highlighted, but the trends observed suggest forest cover around villages but not in the immediate vicinity (1 km) leads to higher risk of malaria both in the north and in the south. The model including average tree crown cover density within 30 km of villages with no temporal lag provided the best AIC. In the final models with the deforestation variables we therefore included the average tree crown cover density within 30 km of villages in the starting year of the temporal scale for the deforestation variable considered (e.g. 3-year lag in the

model with percent area that experienced forest loss in previous 3 years as the deforestation variable).

**Appendix 1—table 3.** IRR [95% CI] associated with a 1% increase in average tree crown density. Adjusted for the probability of seeking treatment, the spatio-temporal structure of the data, the environmental covariates selected in the model and malaria incidence in the previous 1 and 2 months. See Materials and methods for details.

| Time lag | South | | | North | | |
|---|---|---|---|---|---|---|
| | Buffer radius | | | Buffer radius | | |
| | 1 km | 10 km | 30 km | 1 km | 10 km | 30 km |
| Current year | 1 [0.99; 1.01] | 1.07 [1.04; 1.10] | 1.06 [1; 1.12] | 0.99 [0.97; 1.02] | 1.01 [0.96; 1.05] | 1.10 [0.99; 1.23] |
| Previous 1 year | 1 [0.99; 1.02] | 1.07 [1.05; 1.10] | 1.09 [1.03; 1.15] | 1 [0.97; 1.02] | 1.01 [0.97; 1.06] | 1.12 [0.99; 1.26] |
| Previous 2 years | 1 [0.99; 1.02] | 1.07 [1.05; 1.10] | 1.09 [1.03; 1.16] | 1 [0.98; 1.03] | 1.02 [0.97; 1.06] | 1.10 [0.98; 1.25] |
| Previous 3 years | 1 [0.99; 1.02] | 1.07 [1.04; 1.10] | 1.10 [1.04; 1.16] | 1.01 [0.98; 1.03] | 1.02 [0.97; 1.07] | 1.10 [0.98; 1.24] |

## Deforestation - non-linearities

The IRR effect estimates in *Table 1* and *Figure 3* in the main manuscript assume a linear relationship between deforestation and malaria. *Appendix 1—figure 5* shows a few of these relationships – via their individual contribution $\beta * f(X)$ in equation 2 – when such linearity is not imposed in the GAM models. Although the AIC fit is slightly better when modeling non-linearities, these plots show that the linearity assumption is mostly warranted.

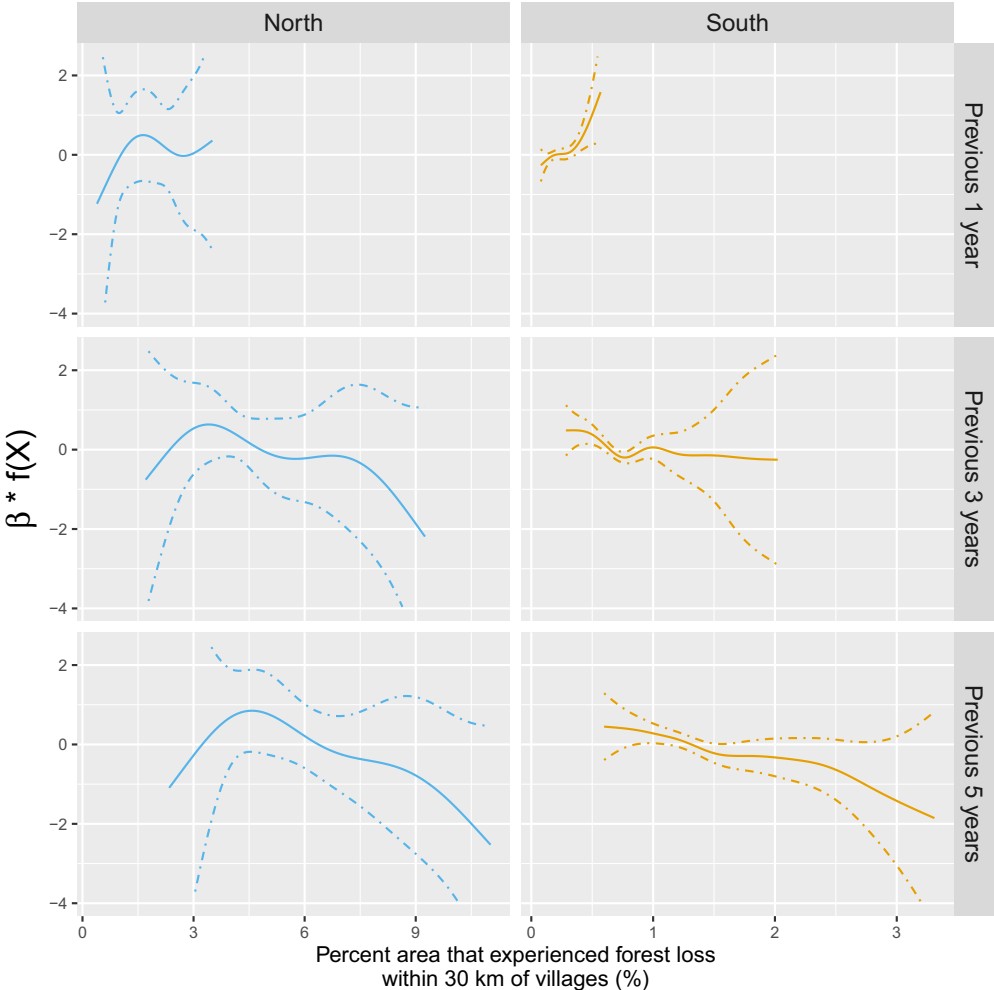

**Appendix 1—figure 5.** Adjusted relationship between deforestation and malaria incidence. All models were adjusted for environmental covariates and forest cover on top of the probability of seeking treatment, the spatio-temporal structure of the data ($f(t)$, $f(Lat, Long)$ and village random intercepts) and malaria incidence in the previous 1 and 2 months. See Materials and methods for details. Note that scales are different between buffer radius for better visualization. *Appendix 1—figure 14* shows the raw scatterplot between monthly village malaria incidence rate and deforestation. *Appendix 1—figure 15* and *Appendix 1—figure 16* show the raw time series of malaria incidence, forest cover and percent area that experienced forest loss.

## S6: AIC fit of the seven monthly climatic variables variations

To avoid collinearity, we have selected (based on the best AIC fit – see *Appendix 1—table 4*) the 1 of the 7 variations (In current month, in previous 1, 2, or 3 months and aggregated over previous 1, 2, or 3 months) of the three monthly climatic variables (Precipitation, Day temperature and Night temperature) to be included in the final model. This was done independently for each of the four outcome models (South, North, South Pf, and South Pv). For South *P. falciparum*, the second best fitting AIC day temperature (in current month) was selected (rather than in previous month) because

of very similar AIC fits and to ensure better comparability with the overall South and South *P. vivax* models.

**Appendix 1—table 4.** AIC fit of univariate models when including each of the seven monthly climatic variation one at a time as unique covariate in equation 2, solely adjusted for the probability of seeking treatment, the spatio-temporal structure of the data ($f(t)$, $f(Lat, Long)$ and village random intercepts).
AIC selected are in bold.

| | Outcome model | | | |
|---|---|---|---|---|
| | **South** | **North** | **South *P. falciparum*** | **South *P. vivax*** |
| Day temperature | | | | |
| Current month | **18,546** | 1671 | **13,226** | **14,575** |
| Previous month | 18,556 | 1702 | 13,224 | 14,590 |
| 2 months ago | 18,578 | **1669** | 13,249 | 14,594 |
| 3 months ago | 18,559 | 1672 | 13,232 | 14,593 |
| Over current and previous month | 18,556 | 1670 | 13,231 | 14,583 |
| Over current and previous 2 months | 18,570 | 1670 | 13,248 | 14,588 |
| Over current and previous 3 months | 18,573 | 1680 | 13,249 | 14,592 |
| Night temperature | | | | |
| Current month | 18,413 | 1669 | 13,120 | 14,474 |
| Previous month | 18,453 | 1670 | 13,155 | 14,520 |
| 2 months ago | 18,547 | 1673 | 13,231 | 14,576 |
| 3 months ago | 18,581 | 1672 | 13,251 | 14,596 |
| Over current and previous month | 18,296 | 1664 | 13,044 | 14,397 |
| Over current and previous 2 months | 18,263 | 1669 | 13,014 | 14,385 |
| Over current and previous 3 months | **18,262** | **1663** | **13,007** | **14,385** |
| Precipitation | | | | |
| Current month | 18,532 | 1693 | 13,198 | 14,593 |
| Previous month | **18,520** | 1669 | **13,181** | **14,575** |
| 2 months ago | 18,538 | **1658** | 13,207 | 14,594 |
| 3 months ago | 18,579 | 1664 | 13,243 | 14,596 |
| Over current and previous month | 18,570 | 1672 | 13,239 | 14,596 |
| Over current and previous 2 months | 18,543 | 1670 | 13,187 | 14,590 |
| Over current and previous 3 months | 18,555 | 1674 | 13,212 | 14,591 |

## S7: Additional figures

This section presents additional figures mentioned in the text and in the additional results section of appendix 1.

## Forest and environmental variables

*Appendix 1—figures 6*, *Appendix 1—figure 7* and *Appendix 1—figure 8*.

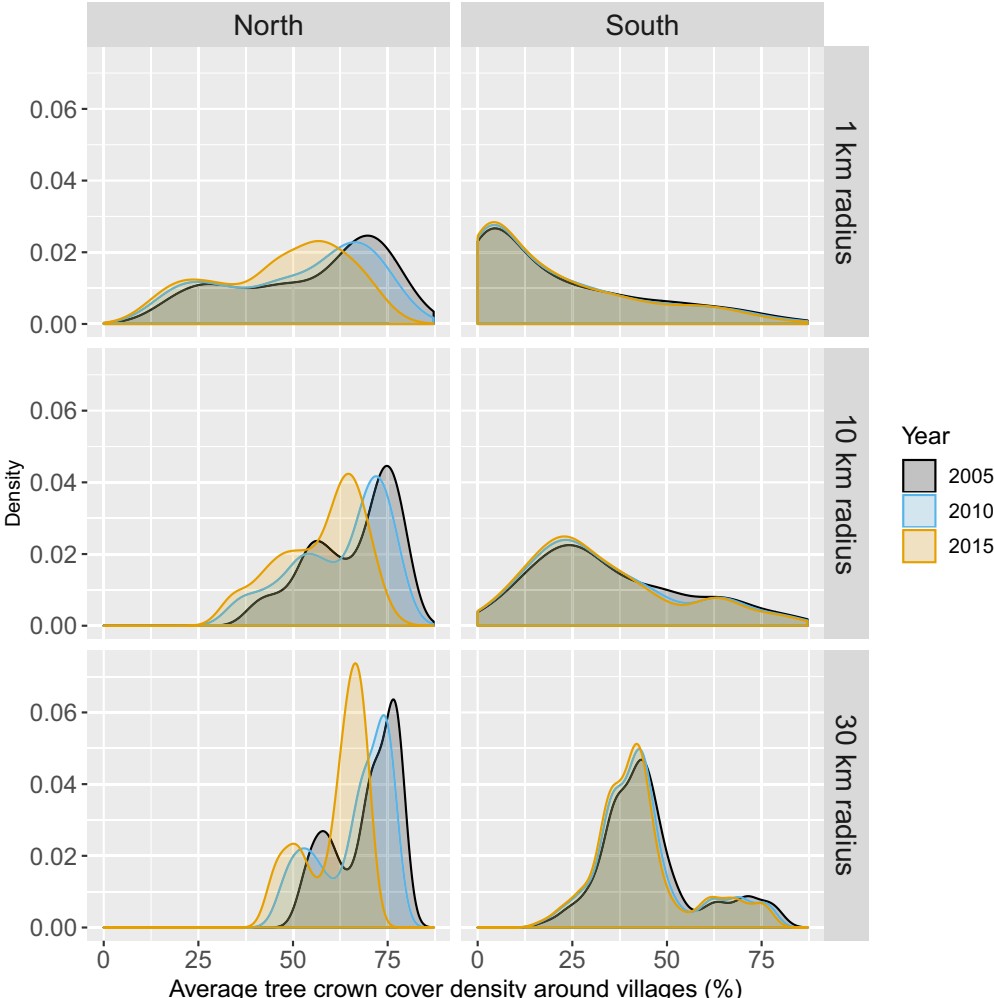

**Appendix 1—figure 6.** Distribution of average tree crown cover density within 1, 10, and 30 km of villages.

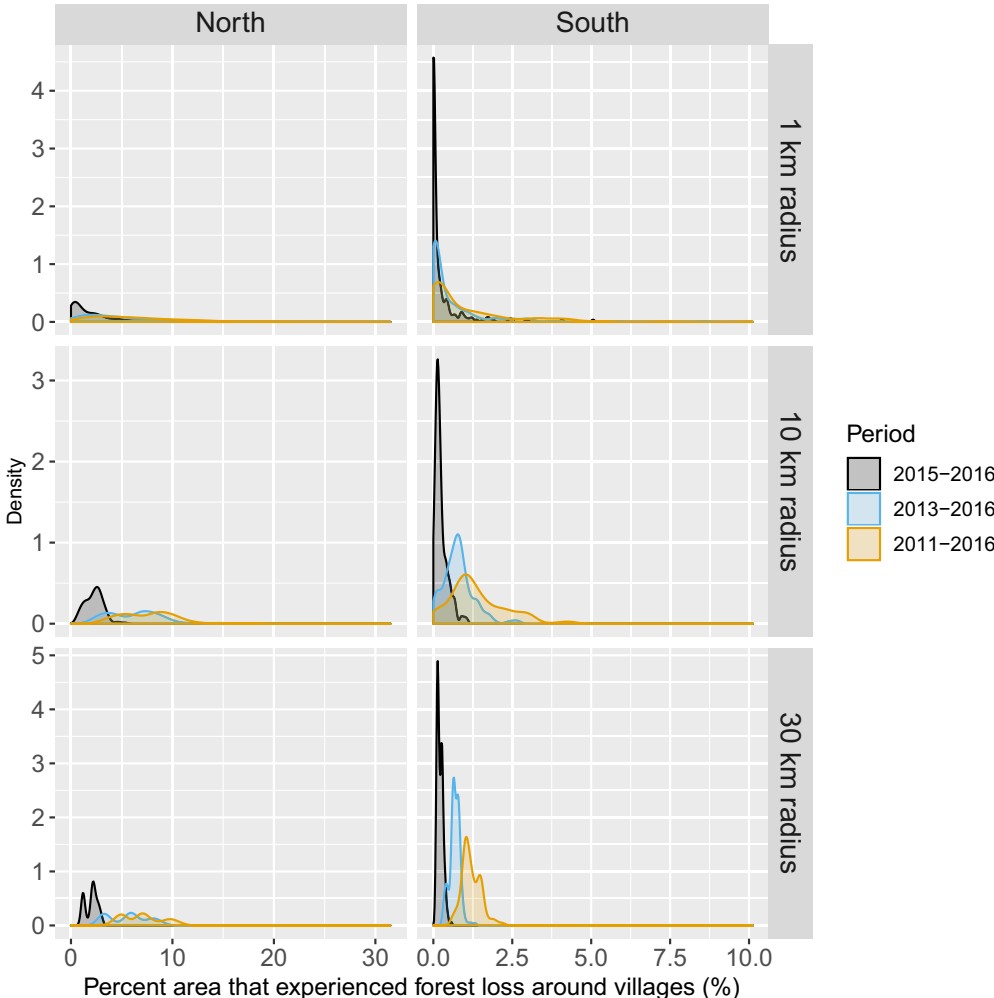

**Appendix 1—figure 7.** Distribution of percent area within 1, 10, and 30 km of villages that experienced forest loss between 2011 and 2016. Note that the scales are different for every panel for better visualization of the distributions.

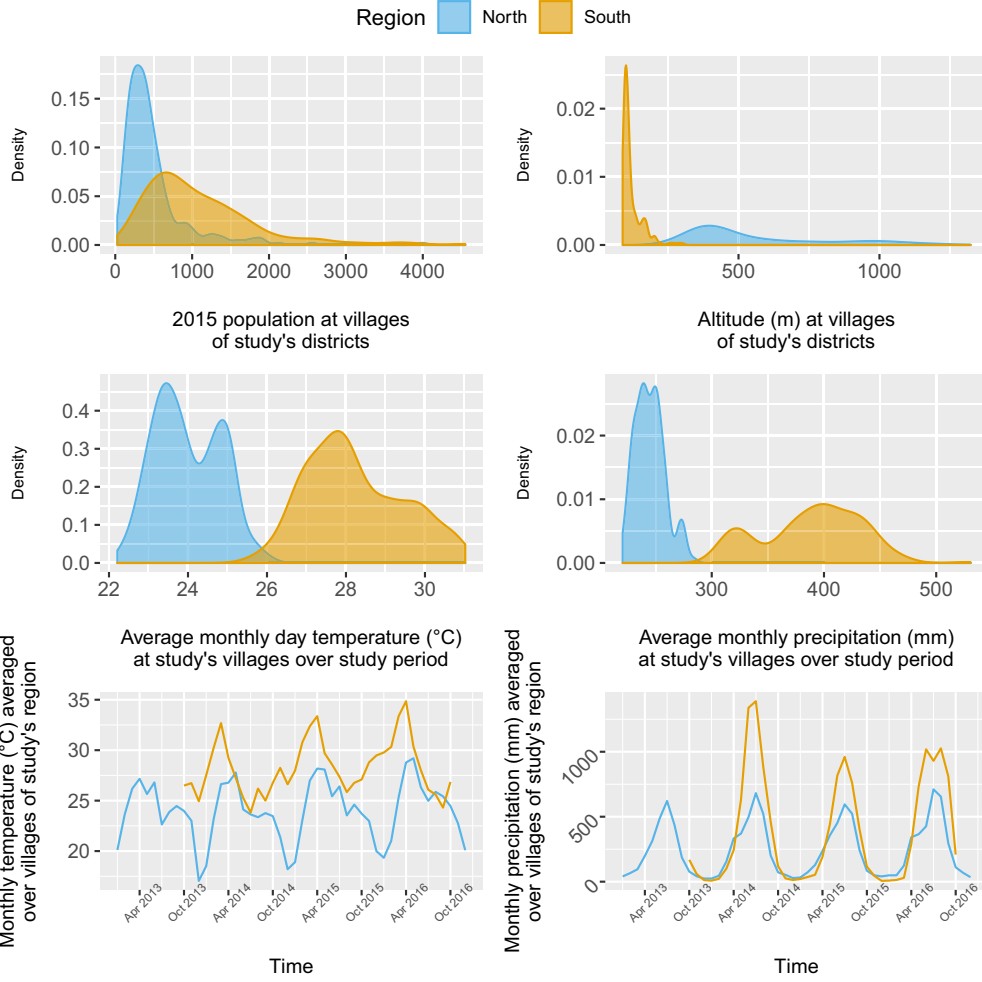

**Appendix 1—figure 8.** Distribution and time series of environmental covariates (population, altitude, monthly day temperature and monthly total precipitation) at study's villages.

## Malaria registries – malaria infections

*Appendix 1—figure 9*.

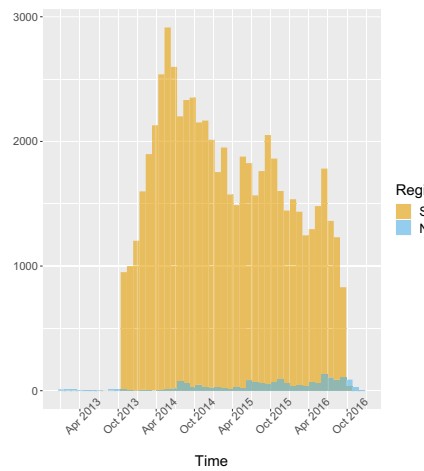

(a) Number of patients recorded per month in health facilities malaria registries over time.

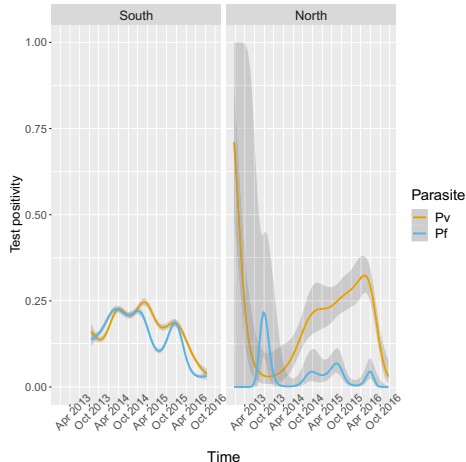

(b) Smoothed (binomial GAM) *P. falciparum* and *P. vivax* test positivity rate over time.

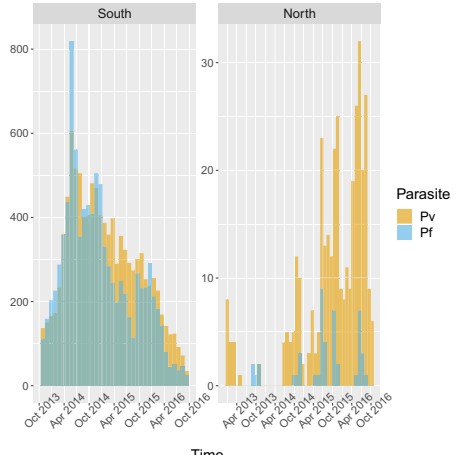

(c) Monthly case count recorded in healh facilities malaria registries. Note that scales are different between regions for better visualization.

**Appendix 1—figure 9.** Additional figures from malaria registries: malaria infections.

## Malaria registries - SES

*Appendix 1—figure 10*.

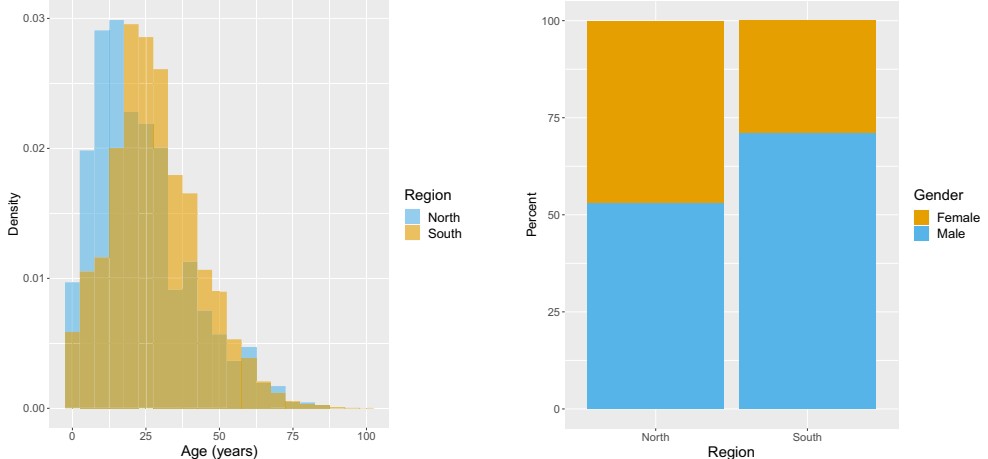

(a) Distribution of age (in years) of patients recorded in the malaria registries.

(b) Distribution of gender of patients recorded in the malaria registries.

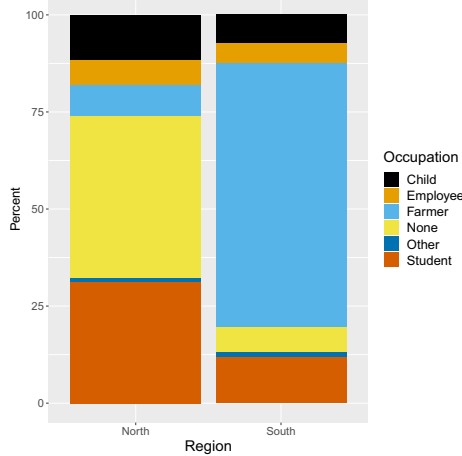

(c) Distribution of occupation of patients recorded in the malaria registries.

**Appendix 1—figure 10.** Distributions of socio-economomical variables of all patients recorded in the malaria registries.

## Malaria registries – matched vs unmatched

*Appendix 1—figure 11.*

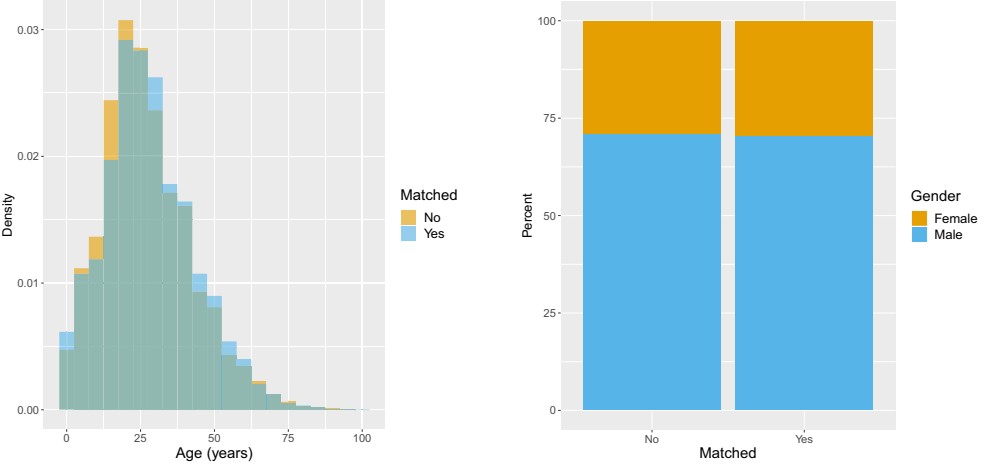

(a) Distribution of age (in years) of patients recorded in the malaria registries.

(b) Distribution of gender of patients recorded in the malaria registries.

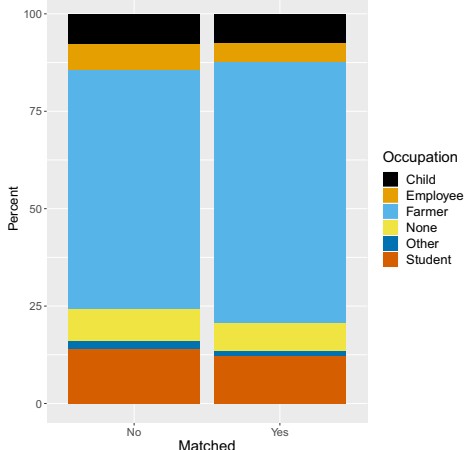

(c) Distribution of occupation of patients recorded in the malaria registries.

**Appendix 1—figure 11.** Additional figures from malaria registries: matched vs unmatched SES variables.

Treatment-seeking

*Appendix 1—figure 12.*

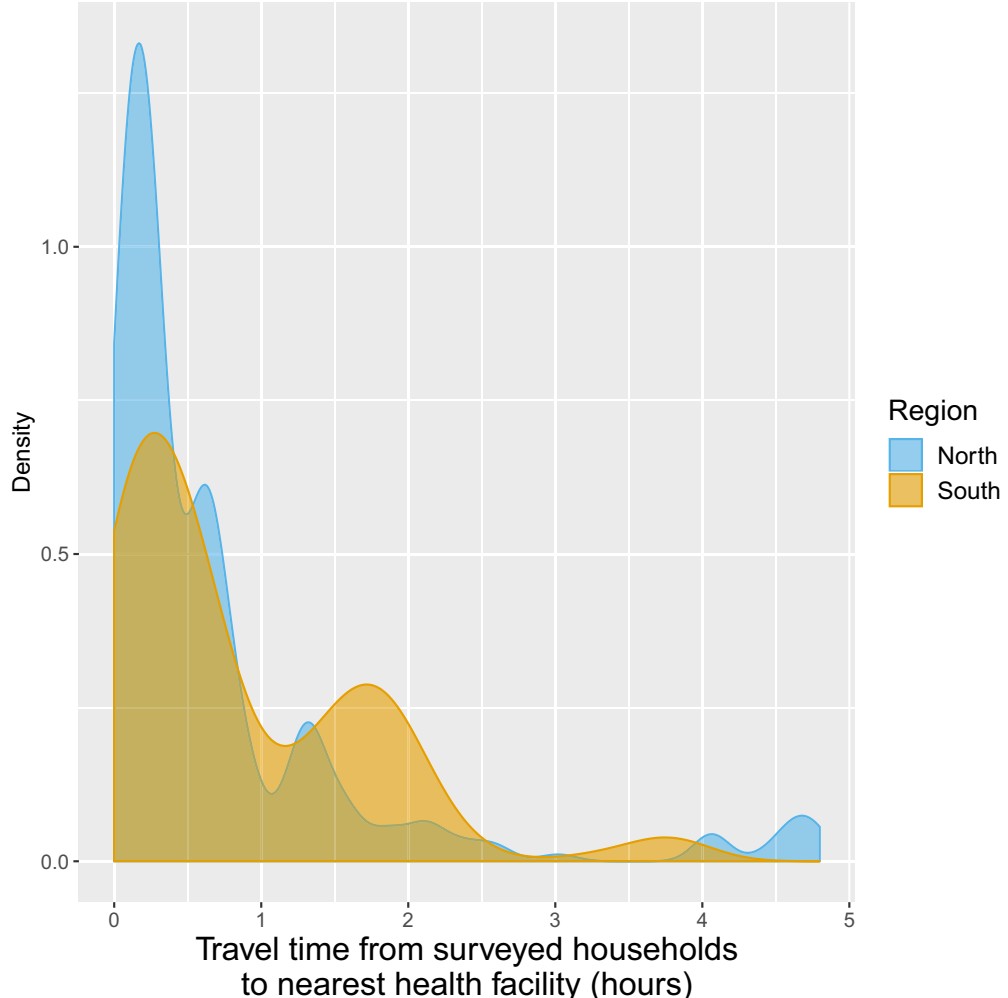

**Appendix 1—figure 12.** Distribution of travel time (in hours) from surveyed households to closest health facilities.

## Statistical analysis – environmental covariates

*Appendix 1—figure 13*.

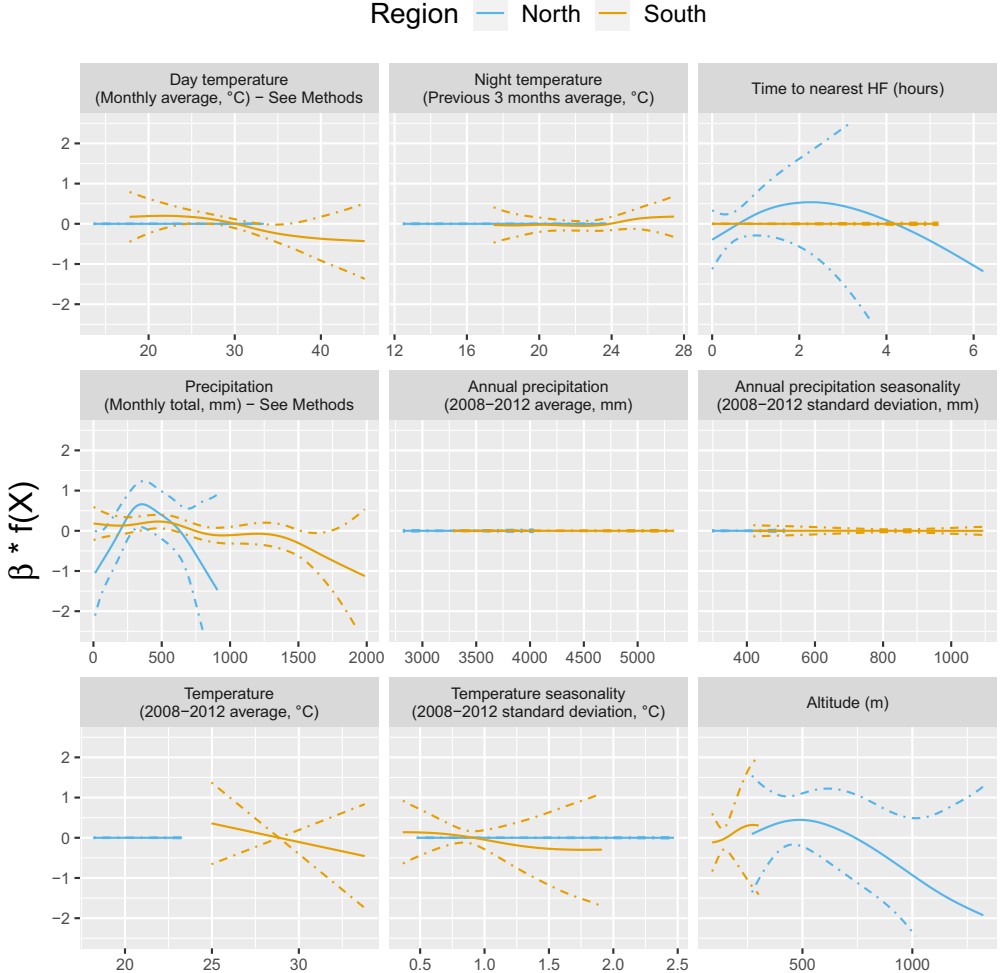

**Appendix 1—figure 13.** Relationships between malaria incidence and the environmental covariates in the multivariable model described in *equation 2* (30 km radius and 1-year temporal lag), additionally adjusted for the probability of seeking treatment, the spatio-temporal structure of the data ($f(t)$, $f(Lat, Long)$ and village random intercepts) and malaria incidence in the previous 1 and 2 months. Dashed lines are for 95% confidence intervals. Note that the y scale has been trimmed a bit for better visualization.

## Raw association between malaria incidence and deforestation

*Appendix 1—figure 14.*

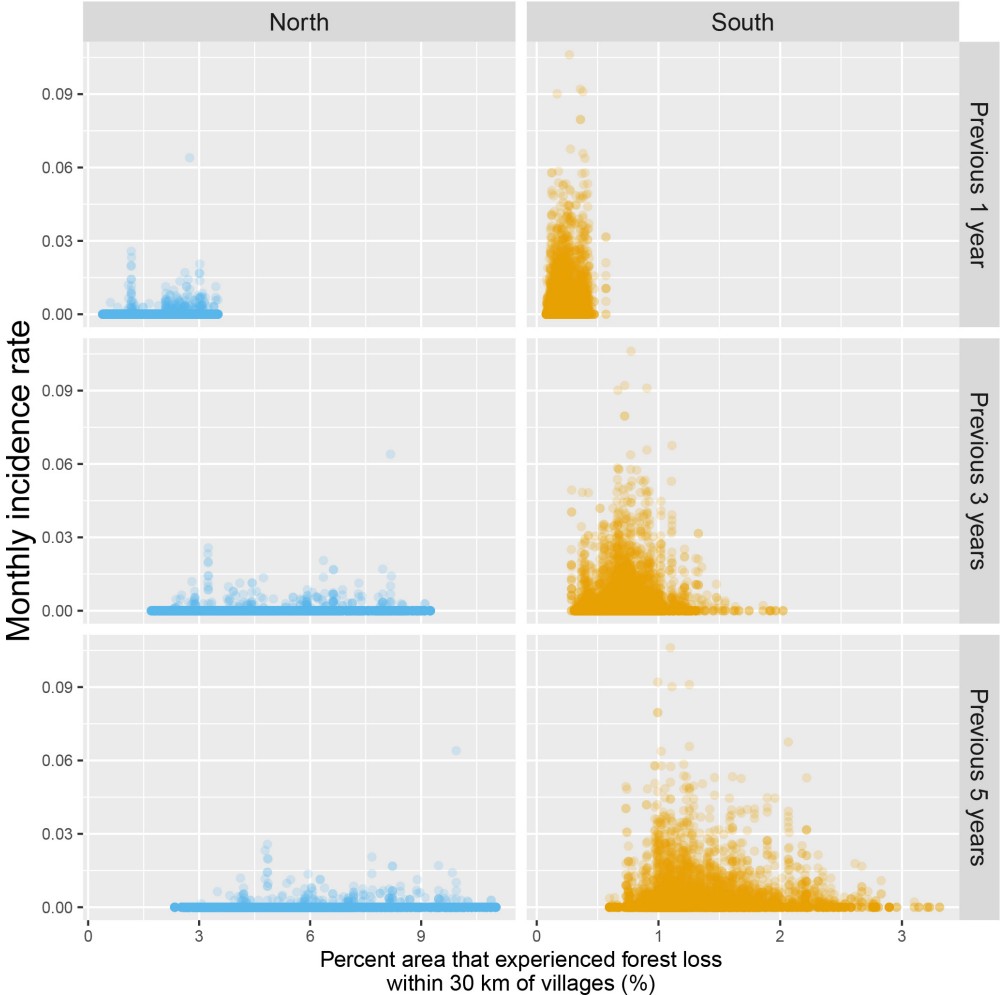

**Appendix 1—figure 14.** Raw scatterplot between monthly village malaria incidence rate and the percent area within 30 km of villages that experienced forest loss in the previous 1, 3, and 5 years. Note that scales are different between regions for better visualization.

Raw time series of malaria incidence, forest cover, and deforestation

*Appendix 1—figure 15* and *Appendix 1—figure 16*.

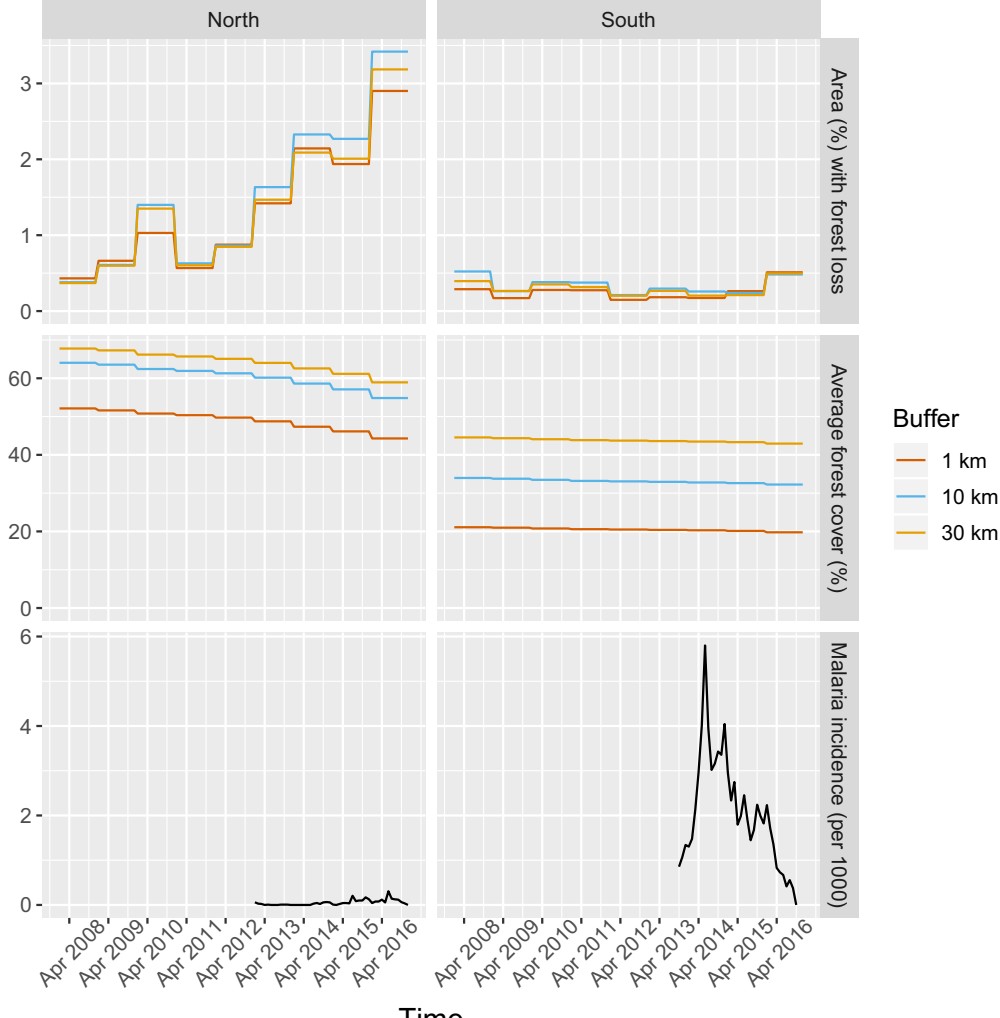

**Appendix 1—figure 15.** Time series of deforestation (percent area that experienced forest loss around villages), forest cover (average tree crow cover around villages) and malaria incidence, averaged over study's villages and for varying buffer radius around villages (1, 10, and 30 km).

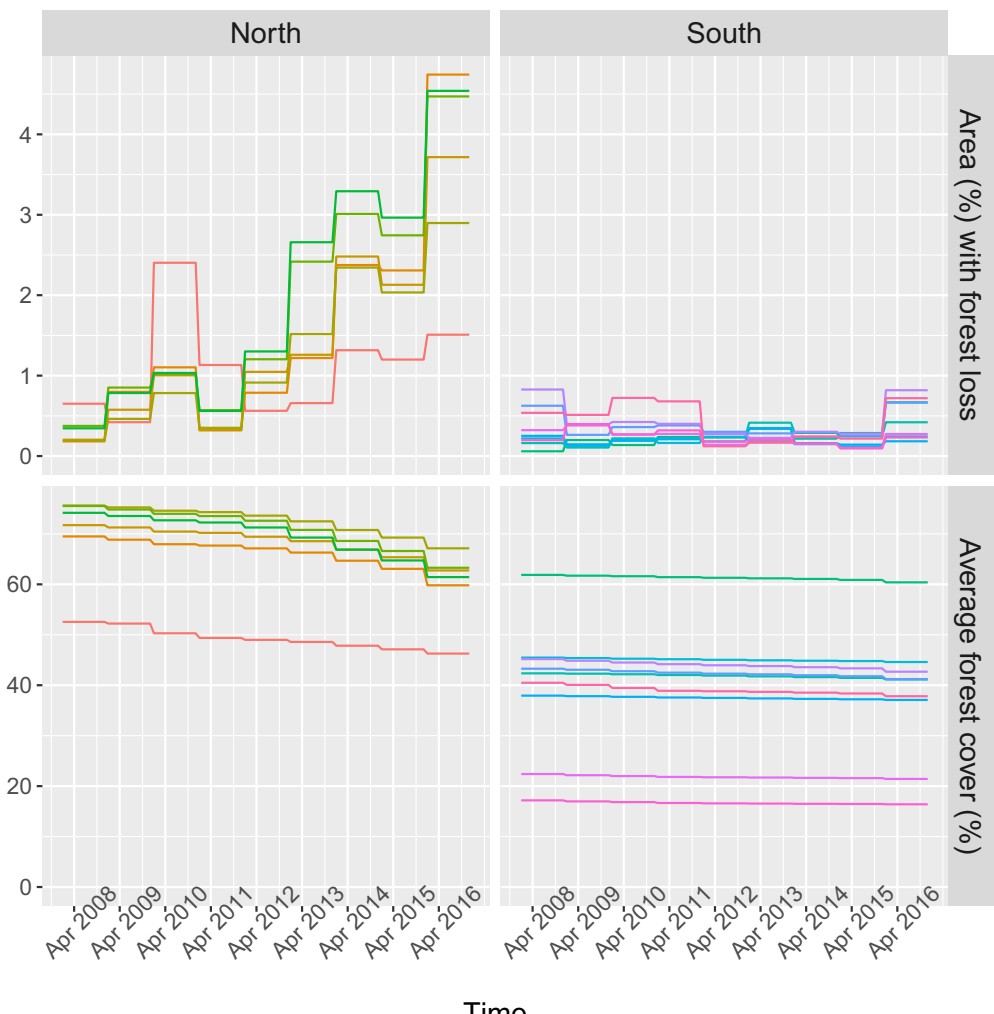

**Appendix 1—figure 16.** Time series of deforestation (percent area that experienced forest loss within 30 km of villages) and forest cover (average tree crown cover within 30 km of villages), for a few randomly sampled study's villages. Each color represents one village.

Statistical analysis – *P. falciparum* and *P. vivax*

*Appendix 1—figure 17.*

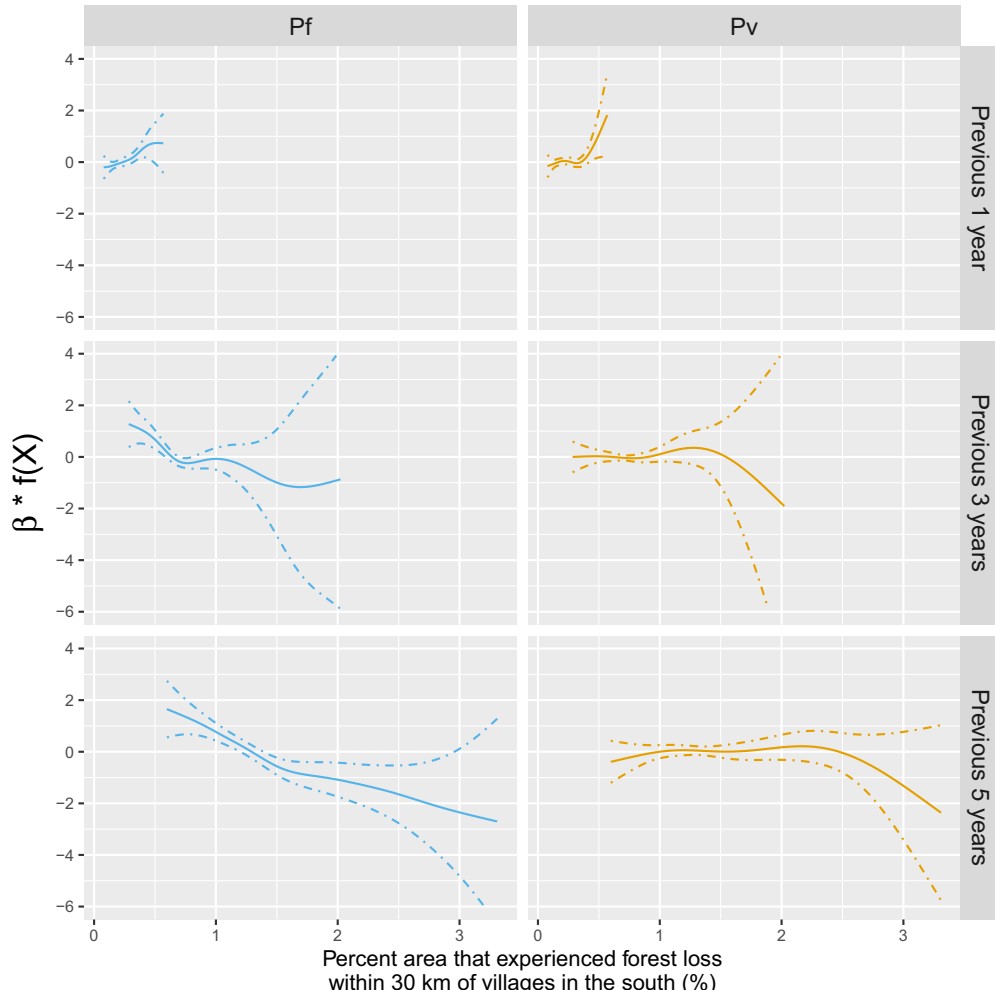

**Appendix 1—figure 17.** Adjusted relationship between deforestation and species-specific malaria incidence in southern Lao PDR. All models were adjusted for environmental covariates and forest cover on top of the probability of seeking treatment, the spatio-temporal structure of the data ($f(t)$, $f(Lat, Long)$ and village random intercepts) and malaria incidence in the previous 1 and 2 months.

