## [Decision Letter]

Thank you for submitting your article "Spatio-temporal associations between deforestation and malaria incidence in Lao PDR" for consideration by *eLife*. Your article has been reviewed by three peer reviewers, and the evaluation has been overseen by a Reviewing Editor and a Senior Editor. The following individual involved in review of your submission has agreed to reveal their identity: James A Watson (Reviewer #1).

The reviewers have discussed the reviews with one another and the Reviewing Editor has drafted this decision to help you prepare a revised submission. Please submit a revised version that addresses these concerns directly.

Summary:

Deforestation is widely thought to have played an important role in the declining incidence of malaria in the Greater Mekong Sub-region but detailed analyses have been lacking. In this paper the authors assemble a large dataset of geolocated malaria cases from Lao PDR together with forest cover data. Using a spatio-temporal modelling framework, they find no evidence of an association of deforestation within 1 or 10km of a village and malaria incidence. However, deforestation within 30km of a village was associated with malaria incidence, though with the direction of the association depended on the timescale considered. In the short term (1-2 years) deforestation was associated with increases in malaria incidence in the south, though no association was found in the north. In the long-term (3-5 years) deforestation was associated with a lower malaria incidence rate, with decreases of about 5% in both the south and the north.

Essential revisions:

1) All reviewers had concerns about variable selection and a number of suggestions were made, including regularisation, the need to adjust for climate anomalies and malaria control programme activities. The differences between the north and south were also highlighted and, following the consultation process, there was a consensus that lumping both parasites together and using one region did not seem appropriate. There were also questions about the choice of initial climate covariates. These issues need to be addressed either by conducting the analysis in a different way (we leave it to the authors to decide on the best way) or by convincing the reviewers that such changes are not needed. A better explanation and consideration of the climate covariates that were considered is also needed.

2) There needs to be a more thorough exploration of the effect of inclusion of malaria cases in the previous month. As reviewer 3 points out, it is surprising that the effect of malaria cases in the previous month is left as a sensitivity analysis as the autocorrelation effect should be important in a transmission system.

3) The concerns about the way forest data are used (as raised by reviewer 2) need to be addressed. The authors should justify how they have derived canopy cover from a dataset not designed to quantify annual forest canopy cover. This is quite a widely used dataset and the interpretation is unconventional. The authors could either re-analyse these data or provide further explanation of how they have modelled canopy cover levels.

4) The authors should include the nonlinear trend f(t) in the results as well as the overall temporal pattern in the regions' deforestation.

5) There needs to be greater clarity about what was done in the results – e.g. regarding the treatment seeking model and adjustment for confounders.

6) There is a need for clarification of differences in malaria regimes in north and south in the Introduction.

Reviewer #1:

This is a carefully written paper on the complex associations between deforestation and malaria incidence. For someone with no background in the relevant literature it provides an easy to read introduction to previous work and shows how these new results fit with previous reports. On the whole I enjoyed reading it. My main suggestions are cosmetic.

I feel that the authors have done themselves a disservice in the Results section when reporting the observed associations between deforestation and malaria incidence. When I first read it, I had the impression that no adjustment for confounders was made! This does not seem to be mentioned at all in the Results. But in fact the underlying model is quite complex (Figure 8) and the authors use elevation, precipitation and temperature as the possible confounders (if I understood correctly). I would suggest adding a few sentences in the Results to clarify this. Maybe the Discussion could have a sentence or two on any unmeasured/residual confounding? Another really important adjustment is done for treatment seeking behaviour. This is rarely done in similar stats analyses and the authors have put quite a lot of work put into constructing a sophisticated model of treatment seeking (Figure 7B is really informative! Relegating this figure to the supplementary materials is a shame.)

I don't think the Results section even mentions the treatment seeking model. It is an important output and when mentioned in the Results, it should be made clear that the treatment seeking model is built using a different dataset, so there is no doubling dipping when constructing the geospatial model.

I'm not very familiar with GAMs so feel free to ignore this comment: instead of doing forward/backward AIC-based variable selection, could you instead fit the full model with appropriate regularisation (ridge regression type but in a GAM framework)?

Final comment concerns *P. vivax*. It is to be expected that the short term associations are dampened as probably the majority of infections are relapses (as mentioned in the Discussion). So you could argue that seeing a dampened association provides some kind of reassurance that there isn't massive unmeasured confounding going on (I would worry if larger associations were seen for vivax). This argument doesn't really work for the long term associations though. Any reasons for why no long term associations are observed?

Reviewer #2:

This study addresses an important research gap in understanding the ecology of malaria within Lao PDR and assembles an impressive dataset of geolocated malaria cases. However, there are major limitations in this study which question the validity of results and the interpretation of data sources.

First, without adjusting for precipitation or temperature differences annually, it is impossible to interpret whether the time lags between deforestation and malaria incidence reflect ecological changes and forest loss or whether this is due to interannual variation in other environmental factors. This time period includes a major El Nino event with extensive forest fires reported across Southeast Asia, including in Lao PDR (e.g. https://www.ncbi.nlm.nih.gov/pmc/articles/PMC6520341/). Without adjusting for the climate anomalies during this time period, it is difficult to attribute changes in risk to deforestation. Additionally, this analysis would ideally include some estimates of malaria control program activities during these time periods and whether this also varied either temporally or spatially. The authors address bias in health system reporting but no other control activities.

Additionally, assuming that the authors are simply using the forest data obtained from Hansen et al., this is not analyzed in a method consistent with how it was produced. This dataset includes tree crown coverage for 2000 (as a percentage) and a forest loss layer which reflects the change in state of a pixel from forest (defined as 50% canopy cover) to non-forest (less than 50%). As the forest loss layer only reflects a change in state (this could be 100% canopy cover to 0% or 52% to 49%), this is can only be interpreted by classifying the initial forest layer using a 50% threshold and cannot be used to estimate canopy cover percentages. There is also a forest gain layer (reflecting the change of state from non-forest to forest) which is not discussed at all within this paper. Following discussions about the accuracy of these datasets, there have also been updates to this data and additional layers available on World Resources Institute which could significantly improve this classification. As well, as forest configuration and fragmentation has widely been associated with malaria incidence in forest settings, this work would be improved by including additional metrics reflecting patterns of deforestation rather than just total area within a circular buffer.

Reviewer #3:

This manuscript addresses an important question in the ecology of malaria, namely what is the impact of deforestation on malaria incidence. Although there have been multiple studies and reviews on the subject for other regions especially the Amazon, an overall picture remains elusive because of the complexity of an environmental impact that is non-stationary in time as proposed in the hypothesis of "frontier malaria". Statistical analyses most often lack the longitudinal data needed to disentangle effects of deforestation as a function of time, and to also consider the spatial scales at which such effects are manifested. The work of Castro and colleagues for the Amazon did consider the temporal axis and provided evidence for a transient increase followed by a decrease to low endemic malaria.

Here, the authors take advantage of a longitudinal and spatially-resolved malaria data set in Lao People's Democratic Republic, together with a forest data set from remote sensing, to investigate how deforestation influences malaria incidence by both *P. falciparum* and *P. vivax* at different spatial and temporal scales. They contrast results for two regions within Lao PDR, North and South, with distinct transmission regimes and populations. They provide evidence consistent with the "frontier malaria" hypothesis, albeit with shorter times for the turn-around, from an increasing to a decreasing trend, than reported for the Amazon. The positive and short-term positive effect on malaria incidence is stronger in the South than in the North, and for *P. falciparum*. These results are important in demonstrating the need to consider statistical analyses that carefully introduce time since deforestation, and in supporting a positive transient effect of deforestation on malaria that should inform both public health strategies and forest management.

I have some comments that should be addressed to make the analyses and their interpretation clearer:

1) The way climate covariates are incorporated may ultimately "work" to capture a complex set of effects in a statistical model, but it is not very convincing. In particular, model selection is conducted on the basis of non-specific cases and the South only. The South and North show very different transmission regimes not just because of different population sizes but because of different environments, especially altitude (and therefore temperatures). In the more endemic regime of the South, malaria cases are likely to reflect much less clear effects of climate covariates than in the low transmission, epidemic, North. In addition, aggregating cases for the different parasites is problematic as their transmission is affected by climate covariates in different ways, given the relapses of *P. vivax* which typically lead to a different seasonality and less sensitivity to climate drivers. *P. falciparum* shows a more epidemic behavior in the North as expected for higher altitudes/cooler temperatures. These differences make one wonder whether it is valid to conduct model selection first in terms of the South and for both parasites together, and then fix these variables in the rest of the analysis.

2) The choice of initial climate covariates is also somewhat confusing. Why use WorldClim at all? One can obtain the means, or totals, and the CVs, from the time series used for the monthly data. WordClim uses a particular window of time (decades) to obtain these "typical" variables. It is my understanding that this time period is earlier than that considered here. Why use altitude and temperature? Aren't these strongly correlated within a region? An alternative consideration is to separate interannual and seasonal effects, by considering monthly values (as done here) and also the mean temperatures (of that same data set) over a window of time that precedes the transmission season (for *P. falciparum*) and corresponds to the rainy season.

3) The effect of malaria cases in the previous month is left for the sensitivity analysis as one variation of the main results. This is surprising as this autocorrelation effect should be important in a transmission system. That is, it would be natural to start with that model and see what is the significance of that variable. Its inclusion seems to do more than just weaken slightly the results. The effect in the North for *P. falciparum* becomes non-significant and that in the South weakens considerably. I wonder what would have been the result of model selection to start with, if this variable had been included. (I understood that the "adjusted" model does not repeat model selection).

4) The distinction of the malaria regimes in the North and South should be described more clearly in the Introduction for a general reader not familiar with malaria dynamics. The North appears seasonal low transmission or epidemic (given the altitudes and the time series), and the South, seasonal endemic. The differences in transmission characteristics between parasites should also be included early in the text; relapses are mentioned late in the Discussion. These considerations are important to how one looks at the time series and results.

5) The nonlinear trend f(t) is mentioned in the model description but not in the results. I could not tell whether deforestation itself exhibits a trend and how consideration or not of f(t) influences the results. Including the resulting f(t) as well as the overall temporal pattern in the regions' deforestation would be informative.

6) Similarly, I would have liked to see in the supplement the main results underlying variable selection and model selection.

---

## [Author Response]

Essential revisions:1) All reviewers had concerns about variable selection and a number of suggestions were made, including regularisation, the need to adjust for climate anomalies and malaria control programme activities. The differences between the north and south were also highlighted and, following the consultation process, there was a consensus that lumping both parasites together and using one region did not seem appropriate. There were also questions about the choice of initial climate covariates. These issues need to be addressed either by conducting the analysis in a different way (we leave it to the authors to decide on the best way) or by convincing the reviewers that such changes are not needed. A better explanation and consideration of the climate covariates that were considered is also needed.

We realize that our model selection approach could be improved based on the reviewers’ suggestions. We have now conducted this analysis in a different way by integrating model selection into the model fitting step (using regularization) without first lumping parasites or regions together.

First, we have selected the best fitting (based on AIC) of the 7 variations (In current month, in previous 1, 2 or 3 months and aggregated over previous 1, 2 or 3 months) for the 3 monthly climatic variables (Precipitation, Day temperature and Night temperature) independently for each of the 4 outcome models (South, North, South Pf and South Pv) to be included for further model selection. This change is described in the Materials and methods section and in Appendix 1—table 4 in supplementary Materials S6 in the appendix.

Second, we have used regularization to integrate model selection into the model fitting step by adding in the GAM an extra penalty to each term so that the coefficients for covariates can be penalized to zero (using the select=TRUE argument in the R mgcv::gam function). We thank the reviewers for this great suggestion, as we agree that it makes our model selection more rigorous and transparent while still offering great comparability between the 4 outcome models (South, North, South Pf and South Pv). This change is described in the Materials and methods section.

Third, we revisited our choice of initial climate variables. As suggested by reviewers, Worldclim variables were removed. Instead, we computed the average and standard deviation of the annual total precipitation and the average monthly temperature for the 2008-2012 years from the monthly time series (CHIRPS and MODIS respectively) already used to extract monthly climatic variables. The 2008-2012 period corresponds to the 5-year time period directly before our malaria data (2013-2016). Including the effect of “long-term” climatic variables in our model, on top of the “short-term” seasonal effect modeled via the monthly climatic variables, is important to capture the spatial differences in overall climate between the villages of our study area. This change is described in the Materials and methods section.

Fourth, we modified the result section to clarify that all of our analyses were controlled for climate variables. Previously, that was only mentioned in the labels of the tables and plots in the result section. We thank the reviewers for pointing out this important caveat in our original manuscript. This change is described in the Results section. In addition, and as suggested by reviewer 1, we also moved the figure of our conceptual model from the appendix into the Materials and methods section (Figure 8) to stress that all of our models were adjusted for environmental variables.

Finally, the reviewers are right to mention the importance of adjusting for malaria control program activities. When designing the study, in collaboration with the national control program, we purposefully excluded regions where we knew large programmatic activities were being implemented. Unfortunately though, there are no good data on coverage of other control program activities in these villages over time and we couldn’t formally adjust for it. That said, to be of importance, this residual unmeasured confounding (spatio-temporal variations in malaria control program activities) would need to be associated with deforestation. We have no reason to believe so, except maybe because of an area’s remoteness. We tried to adjust for this remoteness by including travel time to the closest health facility in the model. We included this additional information in the Materials and methods section. Additionally, we included a sentence about unmeasured residual confounding in the Discussion.

2) There needs to be a more thorough exploration of the effect of inclusion of malaria cases in the previous month. As reviewer 3 points out, it is surprising that the effect of malaria cases in the previous month is left as a sensitivity analysis as the autocorrelation effect should be important in a transmission system.

The reviewers are right to raise that point. In our new analysis, we included malaria cases in the previous month in the model and let regularization penalize it just as any other covariates. In this new analysis, we actually realized that malaria cases in the previous 2 months was necessary as well to fully account for any residual temporal autocorrelation and was therefore also included. Several different forms of the non-linear temporal trend f(t) were also explored (up to 25 spline knots, auto-regressive, cyclic cubic spline) but none accounted for the temporal autocorrelation better. The exploration of the effect of inclusion of malaria cases in the previous 1 and 2 months on residual autocorrelation is now presented in the appendix S4. This change is described in the Materials and methods section.

3) The concerns about the way forest data are used (as raised by reviewer 2) need to be addressed. The authors should justify how they have derived canopy cover from a dataset not designed to quantify annual forest canopy cover. This is quite a widely used dataset and the interpretation is unconventional. The authors could either re-analyse these data or provide further explanation of how they have modelled canopy cover levels.

We thank the reviewers for raising that point and asking for clarifications. This is a formidable dataset and we are grateful to the research team that produced it. We have spent quite some time reviewing documentation in the original article, the supplementary materials of the original article, authors’ replies to published critics of their paper and a Google Earth Engine tutorial for using Hansen et al. global forest cover and change data, developed by the authors themselves.

Reviewer 2 is correct that the dataset includes tree crown coverage for 2000 (as a percentage) and a forest loss layer which reflects the change in state of a pixel from forest (defined as 50% canopy cover) to a non-forest state. On the other hand, our interpretation of a non-forest state differs from what the reviewer suggests as follows:

In the Materials and methods section of Hansen’s paper (see supplementary materials), we read:

“Forest loss was defined as a stand-replacement disturbance. Results were disaggregated by reference percent tree cover stratum (e.g. >50% crown cover to ~0% crown cover) and by year.”

In the authors’ replies to published critics of their original paper, we further read:

“Forest loss was defined as a “stand-replacement disturbance”, meaning the removal or mortality of all tree cover in a Landsat pixel.”

Last, in the tutorial developed by the authors of these data, we read:

“The treecover2000 band […] provides information about the tree cover around the globe in the year 2000. The lossyear band provides information about if and when each 30-meter pixel on the planet was deforested during the study period. Given these two bands, it is possible to create an image of the state of the world's forest at any year in the study period.” Followed by code setting to 0 the tree crown the years after forest loss occurred.

Hence our understanding of the data is that the forest loss layer reflects the change in state of a pixel from forest (defined as 50% canopy cover) to a non-forest state (defined as 0% canopy cover).

With that said, we want to stress that our primary exposure, deforestation, quantifies the occurrence of forest loss rather than its magnitude. As a result, the important distinction discussed above has no impact on our primary exposure. Its impact only pertains to one of our covariates, tree crown cover, the value of which in a given year relies on forest loss leading to a 0% tree crown cover, as depicted in Figure 7 of our manuscript.

We have further replied in detail to reviewer 2’s other comments regarding our use of these data below, when answering their concerns directly. In light of the points raised, we have modified the Discussion to mention these limitations.

4) The authors should include the nonlinear trend f(t) in the results as well as the overall temporal pattern in the regions' deforestation.

We thank the reviewers for these suggestions. We have added Appendix 1—figure 4 in the “Additional Results” supplementary material S5 in the appendix, showing the non-linear trend f(t). Appendix 1—figure 15 and Appendix 1—figure 16 in the “Additional figures” supplementary material S6 in the appendix show the overall temporal pattern in the regions’ deforestation.

5) There needs to be greater clarity about what was done in the results – e.g. regarding the treatment seeking model and adjustment for confounders.

We thank the reviewers for pointing out this important caveat in our original manuscript. We have added a few sentences about the treatment seeking results in the Results section. In addition, and as suggested by reviewer 1, we also moved the figure of our conceptual model from the appendix into the Materials and methods section (Figure 8) to stress that all of our models were adjusted for confounders and highlight the importance of our treatment-seeking model. Last, we included further details in labels of all tables and figures.

6) There is a need for clarification of differences in malaria regimes in north and south in the Introduction.

We thank the reviewers for this important comment. We have modified the Introduction to highlight the different malaria regimes in the North and South. We also included additional reference to the different regional transmission regimes in the Results section.

Reviewer #1:This is a carefully written paper on the complex associations between deforestation and malaria incidence. For someone with no background in the relevant literature it provides an easy to read introduction to previous work and shows how these new results fit with previous reports. On the whole I enjoyed reading it. My main suggestions are cosmetic.I feel that the authors have done themselves a disservice in the Results section when reporting the observed associations between deforestation and malaria incidence. When I first read it, I had the impression that no adjustment for confounders was made! This does not seem to be mentioned at all in the Results. But in fact the underlying model is quite complex (Figure 8) and the authors use elevation, precipitation and temperature as the possible confounders (if I understood correctly). I would suggest adding a few sentences in the Results to clarify this. Maybe the Discussion could have a sentence or two on any unmeasured/residual confounding? Another really important adjustment is done for treatment seeking behaviour. This is rarely done in similar stats analyses and the authors have put quite a lot of work put into constructing a sophisticated model of treatment seeking (Figure 7B is really informative! Relegating this figure to the supplementary materials is a shame.)I don't think the Results section even mentions the treatment seeking model. It is an important output and when mentioned in the Results, it should be made clear that the treatment seeking model is built using a different dataset, so there is no doubling dipping when constructing the geospatial model.

We thank the reviewer for pointing out these important caveats in our original manuscript.

We modified the result section to clarify that all of our analyses were controlled for climate variables. Previously, that was only mentioned in the labels of the tables and plots in the result section. This change is described in the manuscript.

We have also added a few sentences in a new “Treatment seeking” section in the Results section. We appreciate the reviewer’s enthusiasm and appreciation for that piece of work. We present it mainly in the appendix not to distract from the core analysis.

In addition, and as suggested below, we also moved the figure of our conceptual model from the appendix into the Materials and methods section (now Figure 8) to stress that all of our models were adjusted for confounders and highlight the importance of our treatment-seeking model. Last, we included further details in labels of all tables and figures.

Finally, we included a sentence about unmeasured residual confounding in the Discussion.

I'm not very familiar with GAMs so feel free to ignore this comment: instead of doing forward/backward AIC-based variable selection, could you instead fit the full model with appropriate regularisation (ridge regression type but in a GAM framework)?

The response to this comment is addressed in the third paragraph of our response to the editor’s essential revision comment #1.

Final comment concerns P. vivax. It is to be expected that the short term associations are dampened as probably the majority of infections are relapses (as mentioned in the Discussion). So you could argue that seeing a dampened association provides some kind of reassurance that there isn't massive unmeasured confounding going on (I would worry if larger associations were seen for vivax). This argument doesn't really work for the long term associations though. Any reasons for why no long term associations are observed?

We thank the reviewer for this constructive feedback. As mentioned in the Discussion, another recent study in the Amazon found similar attenuation of effects for vivax and we agree with the reviewer’s rational.

We think the same argument works for the long-term associations. Vivax long term infections, that can remain for years, when relapsing attenuate the decrease in incidence associated with deforestation in the long-term and pull the estimates towards 1. A study looking at additional larger long-term time lags for vivax might help.

Reviewer #2:This study addresses an important research gap in understanding the ecology of malaria within Lao PDR and assembles an impressive dataset of geolocated malaria cases. However, there are major limitations in this study which question the validity of results and the interpretation of data sources.First, without adjusting for precipitation or temperature differences annually, it is impossible to interpret whether the time lags between deforestation and malaria incidence reflect ecological changes and forest loss or whether this is due to interannual variation in other environmental factors. This time period includes a major El Nino event with extensive forest fires reported across Southeast Asia, including in Lao PDR (e.g. https://www.ncbi.nlm.nih.gov/pmc/articles/PMC6520341/). Without adjusting for the climate anomalies during this time period, it is difficult to attribute changes in risk to deforestation.

We thank the reviewer for raising that point and asking for clarifications. As raised by the other reviewers, it appears clearly that our original manuscript failed to convey the key information that all of our models were adjusted for environmental covariates, including precipitation and temperature.

We modified the result section to clarify that all of our analyses were controlled for climate variables including altitude as well as short term (monthly) and long-term (5-year average) precipitation and day and night temperature. Previously, that was only mentioned in the labels of the tables and plots in the result section. This change is described in the Results section. Additionally, and as suggested by reviewer 1, we moved the figure of our conceptual model from the appendix into the Materials and methods section (now Figure 8) to stress that all of our models were adjusted for confounders. Last, we included further details in labels of all tables and figures.

The reviewer is right to point out that our study does not distinguish between different causes of deforestation. We thank the reviewer for sharing additional reference, bringing to light the importance of natural fires in the GMS. The paper actually shows no particular changes in the occurrence of forest fires in Lao PDR over the study period, but we have to acknowledge that the Hansen forest data used does not distinguish man-made from natural causes of deforestation. We included that limitation in the Discussion.

Additionally, this analysis would ideally include some estimates of malaria control program activities during these time periods and whether this also varied either temporally or spatially. The authors address bias in health system reporting but no other control activities.

The response to this comment is addressed in the sixth paragraph of our response to the editor’s essential revision comment #1.

Additionally, assuming that the authors are simply using the forest data obtained from Hansen et al., this is not analyzed in a method consistent with how it was produced. This dataset includes tree crown coverage for 2000 (as a percentage) and a forest loss layer which reflects the change in state of a pixel from forest (defined as 50% canopy cover) to non-forest (less than 50%). As the forest loss layer only reflects a change in state (this could be 100% canopy cover to 0% or 52% to 49%), this is can only be interpreted by classifying the initial forest layer using a 50% threshold and cannot be used to estimate canopy cover percentages.

The response to this comment is addressed in our response to the editor’s essential revision comment #3.

There is also a forest gain layer (reflecting the change of state from non-forest to forest) which is not discussed at all within this paper.

The reviewer is right to mention the forest gain layer, present in the data, but not used nor mentioned in our paper.

Unlike the forest loss layer, which indicates the year in which forest loss occurred, the forest gain layer is simply a binary variable that takes the value 1 if forest gain (“defined as the inverse of loss or the establishment of tree cover from a non-treed state within a Landsat pixel”, as described in the authors’ replies) occurred between 2000 and 2017 and 0 if not. That is, we don’t know when it happens. In addition of lacking temporal resolution, the forest gain layer does not specify the percent tree crown cover reached after forest gain occurred. One of our key covariates, tree crown cover, would have benefited from information on forest gain but the layer present in the data is not informative enough.

On the other hand, our primary exposure, deforestation, is about the occurrence of forest loss which is fully quantified in the forest loss layer and doesn’t need the forest gain layer.

With that said, the assumptions behind our forest variables, presented in Figure 7, and related to both our understanding of the forest loss layer and the non-use of the forest gain layer but also the raw data’s limitation are key, and we thank the reviewer for asking for clarifications. We have modified the Discussion to mention this limitation.

Following discussions about the accuracy of these datasets, there have also been updates to this data and additional layers available on World Resources Institute which could significantly improve this classification. As well, as forest configuration and fragmentation has widely been associated with malaria incidence in forest settings, this work would be improved by including additional metrics reflecting patterns of deforestation rather than just total area within a circular buffer.

We have used the latest dataset available at the time (Summer 2019) which was an updated version of the dataset, first published in 2013. The reviewer is correct to mention alternative definitions of deforestation such as forest configuration and fragmentation, which have been used in some of the references cited in our paper. We understand that adding other definitions of deforestation would strengthen our work, but we feel this goes beyond the scope of our analysis, which is primarily focused on the exploration of associations over varying spatial and temporal scales. Also, please note that in Table 3 and Figure 5, we have explored one alternative definition to our primary deforestation exposure, restricting deforestation to the occurrence of forest loss in densely forested area.

Reviewer #3:This manuscript addresses an important question in the ecology of malaria, namely what is the impact of deforestation on malaria incidence. Although there have been multiple studies and reviews on the subject for other regions especially the Amazon, an overall picture remains elusive because of the complexity of an environmental impact that is non-stationary in time as proposed in the hypothesis of "frontier malaria". Statistical analyses most often lack the longitudinal data needed to disentangle effects of deforestation as a function of time, and to also consider the spatial scales at which such effects are manifested. The work of Castro and colleagues for the Amazon did consider the temporal axis and provided evidence for a transient increase followed by a decrease to low endemic malaria.Here, the authors take advantage of a longitudinal and spatially-resolved malaria data set in Lao People's Democratic Republic, together with a forest data set from remote sensing, to investigate how deforestation influences malaria incidence by both *P. falciparum* and *P. vivax* at different spatial and temporal scales. They contrast results for two regions within Lao PDR, North and South, with distinct transmission regimes and populations. They provide evidence consistent with the "frontier malaria" hypothesis, albeit with shorter times for the turn-around, from an increasing to a decreasing trend, than reported for the Amazon. The positive and short-term positive effect on malaria incidence is stronger in the South than in the North, and for *P. falciparum*. These results are important in demonstrating the need to consider statistical analyses that carefully introduce time since deforestation, and in supporting a positive transient effect of deforestation on malaria that should inform both public health strategies and forest management.I have some comments that should be addressed to make the analyses and their interpretation clearer:1) The way climate covariates are incorporated may ultimately "work" to capture a complex set of effects in a statistical model, but it is not very convincing. In particular, model selection is conducted on the basis of non-specific cases and the South only. The South and North show very different transmission regimes not just because of different population sizes but because of different environments, especially altitude (and therefore temperatures). In the more endemic regime of the South, malaria cases are likely to reflect much less clear effects of climate covariates than in the low transmission, epidemic, North. In addition, aggregating cases for the different parasites is problematic as their transmission is affected by climate covariates in different ways, given the relapses of P. vivax which typically lead to a different seasonality and less sensitivity to climate drivers. *P. falciparum* shows a more epidemic behavior in the North as expected for higher altitudes/cooler temperatures. These differences make one wonder whether it is valid to conduct model selection first in terms of the South and for both parasites together, and then fix these variables in the rest of the analysis.

The response to this comment is addressed in the first three paragraphs of our response to the editor’s essential revision comment #1.

(2) The choice of initial climate covariates is also somewhat confusing. Why use WorldClim at all? One can obtain the means, or totals, and the CVs, from the time series used for the monthly data. WordClim uses a particular window of time (decades) to obtain these "typical" variables. It is my understanding that this time period is earlier than that considered here. Why use altitude and temperature? Aren't these strongly correlated within a region? An alternative consideration is to separate interannual and seasonal effects, by considering monthly values (as done here) and also the mean temperatures (of that same data set) over a window of time that precedes the transmission season (for *P. falciparum*) and corresponds to the rainy season.

The response to this comment is addressed in the fourth paragraph of our response to the editor’s essential revision comment #1.

In addition, the reviewer is right in that temperature and altitude are not independent of each other but the landscape in northern Lao PDR is very craggy and much more variable than temperature. Both of them are included as initial covariates and regularization may penalized them to zero if needed.

(3) The effect of malaria cases in the previous month is left for the sensitivity analysis as one variation of the main results. This is surprising as this autocorrelation effect should be important in a transmission system. That is, it would be natural to start with that model and see what is the significance of that variable. Its inclusion seems to do more than just weaken slightly the results. The effect in the North for *P. falciparum* becomes non-significant and that in the South weakens considerably. I wonder what would have been the result of model selection to start with, if this variable had been included. (I understood that the "adjusted" model does not repeat model selection).

The response to this comment is addressed in our response to the editor’s essential revision comment #2.

(4) The distinction of the malaria regimes in the North and South should be described more clearly in the Introduction for a general reader not familiar with malaria dynamics. The North appears seasonal low transmission or epidemic (given the altitudes and the time series), and the South, seasonal endemic. The differences in transmission characteristics between parasites should also be included early in the text; relapses are mentioned late in the Discussion. These considerations are important to how one looks at the time series and results.

The response to this comment is addressed in our response to the editor’s essential revision comment #6.

5) The nonlinear trend f(t) is mentioned in the model description but not in the results. I could not tell whether deforestation itself exhibits a trend and how consideration or not of f(t) influences the results. Including the resulting f(t) as well as the overall temporal pattern in the regions' deforestation would be informative.

The response to this comment is addressed in our response to the editor’s essential revision comment #4.

6) Similarly, I would have liked to see in the supplement the main results underlying variable selection and model selection.

Regularization has now integrated model selection and model fitting into one step, simplifying our methodology and making it more rigorous and transparent. See Appendix 1—table 4 in supplementary Materials S6 in the appendix for results on the monthly climatic variable selection.